# Endothelial SMAD1/5 signaling couples angiogenesis to osteogenesis in juvenile bone
Annemarie Lang [1,2,11] ✉, Andreas Benn[3,4], Joseph M. Collins[1], Angelique Wolter[2,5], Tim Balcaen[6,7,8], Greet Kerckhofs [6,7,9,10], An Zwijsen [3] & Joel D. Boerckel [1] ✉

Skeletal development depends on coordinated angiogenesis and osteogenesis. Bone morphogenetic proteins direct bone formation in part by activating SMAD1/5 signaling in osteoblasts. However, the role of SMAD1/5 in skeletal endothelium is unknown. Here, we found that endothelial cell-conditional SMAD1/5 depletion in juvenile mice caused metaphyseal and diaphyseal hypervascularity, resulting in altered trabecular and cortical bone formation. SMAD1/5 depletion induced excessive sprouting and disrupting the morphology of the metaphyseal vessels, with impaired anastomotic loop formation at the chondro-osseous junction. Endothelial SMAD1/5 depletion impaired growth plate resorption and, upon long-term depletion, abrogated osteoprogenitor recruitment to the primary spongiosa. Finally, in the diaphysis, endothelial SMAD1/5 activity was necessary to maintain the sinusoidal phenotype, with SMAD1/5 depletion inducing formation of large vascular loops and elevated vascular permeability. Together, endothelial SMAD1/5 activity sustains skeletal vascular morphogenesis and function and coordinates growth plate remodeling and osteoprogenitor recruitment dynamics in juvenile mouse bone.

The development of the skeleton depends on spatiotemporally coordinated blood vessel morphogenesis and bone formation[1–3]. In development, multicellular patterning is coordinated by morphogens[4]. During bone development, morphogens of the bone morphogenetic protein (BMP) family are principal regulators of osteogenesis and signal via intracellular effector proteins, including SMAD1 and SMAD5 (SMAD1/5)[5,6]. Despite the abundance of BMP ligands and the coordinated coupling of angiogenesis and osteogenesis during bone growth, the role of SMAD1/5 signaling in skeletal endothelium has not been studied[7,8].

Diverse BMP ligands are abundant during bone development and are expressed by a variety of cell types, including skeletal endothelial cells[9–14]. The functions of these morphogens, and that of their downstream SMAD signaling, has been studied extensively in skeletal-lineage cells, resulting in FDA-approved therapies for bone formation and regeneration[14–17]. Multiple studies in various established angiogenesis models, including embryonic development, the mouse retina, and the zebrafish[13,18–20], demonstrate that SMAD signaling is also important to endothelial cell function. Previously, we demonstrated that, during embryonic mouse development, SMAD1/5 synergize with Notch signaling to balance the selection of tip and stalk cells in developmental vascular sprouting[21]. Further, we observed that endothelial cell-specific depletion of SMAD1/5 during early postnatal retinal angiogenesis reduced the number of tip cells, caused hyperdensity of the vascular plexus, and induced arteriovenous malformations[22]. However, the role of SMAD1/5 in long-bone blood vessels, which develop in this particularly BMP-rich niche, is unknown.

Endothelial cells exhibit remarkable genetic and phenotypic heterogeneity, which enables diverse and specialized vascular functions. In the juvenile skeleton, specialized metaphyseal vessels run parallel to the trabeculae in the spongiosa exhibiting columnar/looping structures, and functionally couple angiogenesis to osteogenesis at the growth plate[7]. In contrast, diaphyseal vessels have a sinusoidal structure, form by sprouting angiogenesis in the bone marrow, and functionally couple with hematopoiesis. Both metaphyseal and diaphyseal vessels can be targeted for genetic manipulation using Cdh5-Cre[ERT2]-mediated recombination[7].

Here, we performed Cdh5-conditional homozygous depletion of both SMAD1 and SMAD5 to evaluate the role of BMP-SMAD signaling in the formation, maintenance, and function of metaphyseal and diaphyseal vessels and the coupling of angiogenesis to osteogenesis in juvenile bone. We analyzed both short-term (7 days) and long-term (14 days) consequences of SMAD1/5 depletion from the endothelium of both juvenile and adolescent mice. Juvenile (P21-P35) and adolescent (P42-P56) ages were selected to evaluate vascular morphogenesis during periods of rapid and modest bone formation, respectively. We show that endothelial SMAD1/5 signaling regulates both metaphyseal and diaphyseal vessel morphogenesis,

maintenance, and function, and couples angiogenesis to growth plate remodeling and osteoprogenitor cell maintenance in the juvenile bone. These findings provide insights into how endothelial BMP-SMAD1/5 signaling contributes to bone formation and homeostasis and may contribute to a better understanding of clinical applications of BMPs for vascularized bone regeneration[23].

## Results

### SMAD1/5 restricts vessel volume and width during vascular growth in long bones

To study the role of endothelial SMAD1/5 signaling in morphogenesis of the long bone vasculature, we generated inducible, endothelial cell-conditional (Cdh5-Cre[ERT2]) *Smad1/5* double knockout mice (SMAD1/5[iΔEC]), which were compared to Cre-negative littermate controls (SMAD1/5[WT]). First, mice were injected daily with tamoxifen at postnatal day 19–21 (P19P21) and tibia samples were taken at P28 (Fig. 1a). Efficiency of Cre-recombination in ECs was shown previously[22] and verified by reduction in phospho-SMAD1/5/8-positive ECs in the bone marrow (Supplementary Fig. 1). We used contrast-enhanced microfocus X-ray computed tomography (CECT) analysis of the tibia to visualize and quantify the metaphyseal and bone marrow vasculature in 3D. Endothelial cell-conditional SMAD1/5 depletion at weaning resulted in significantly dilated vessels with disrupted morphology in both metaphyseal and diaphyseal vessels within one week post-knockout (Fig. 1b). Specifically, SMAD1/5 depletion

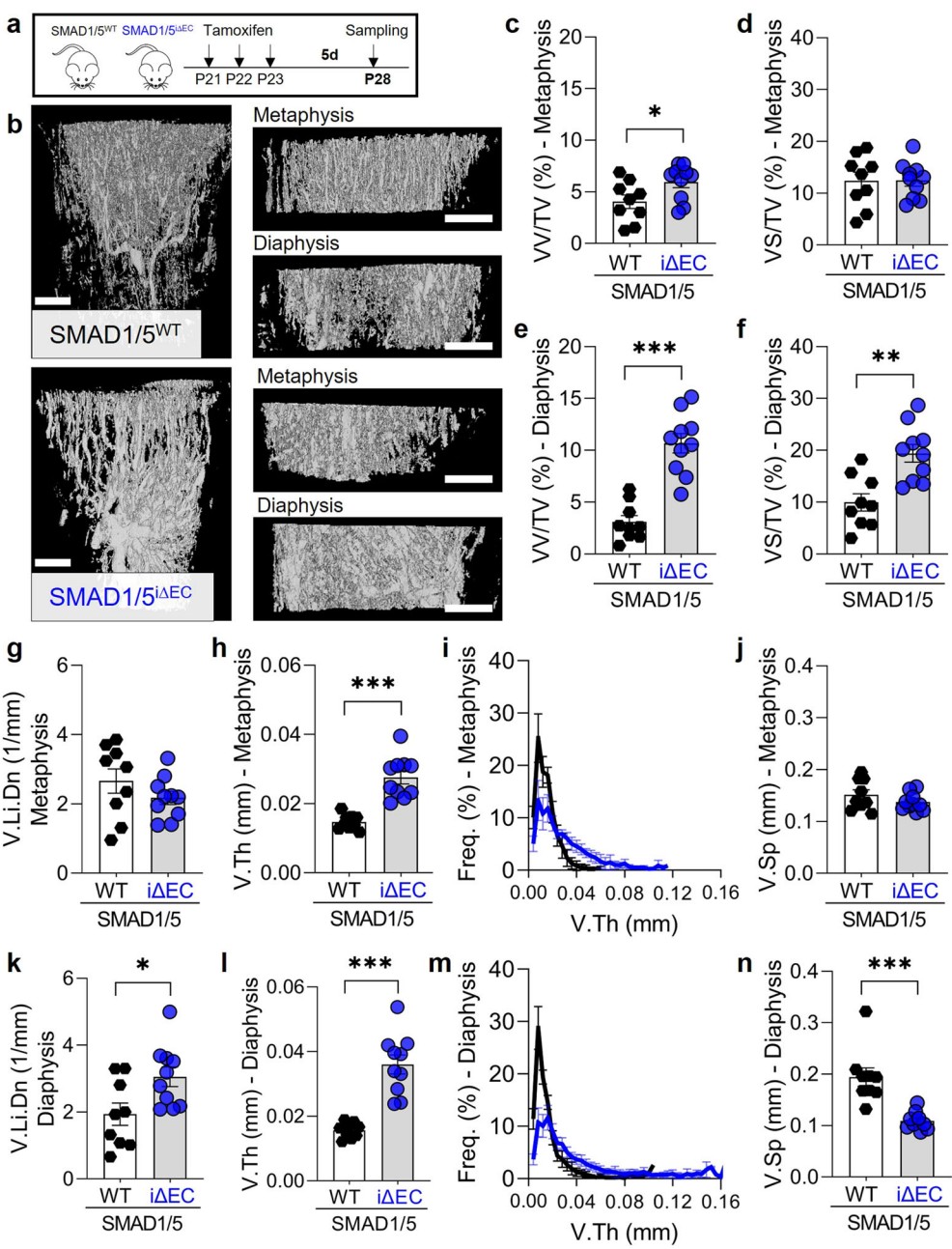

**Fig. 1 | Short-term endothelial SMAD1/5 depletion after weaning increased metaphyseal and diaphyseal vascularity. a** Tamoxifen treatment and short term sampling scheme. Mice were injected postnatal day 19–21 (P19-21) and samples were collected at P28. **b** CECT-based 3D rendering visualizing vessels (P28). Quantitative CECT-based structural analysis (P28; $n^{WT} = 9$; $n^{iΔEC} = 10$) of **c** relative vessel volume (VV/TV) and **d** surface (VS/TV) in the metaphysis or **e, f** diaphysis, respectively. **g, k** vessel linear density (V.Li.Dn), **h, l** mean vessel thickness (V.Th) and **i, m** frequency, as well as **j, n** vascular separation (V.Sp) in metaphysis and diaphysis, respectively. Bar graphs show mean ± SEM and individual data points. Two-sample *t*-test or Mann–Whitney *U*-test (V.Sp, diaphysis) was used to determine the statistical significance; *p*-values are indicated with \**p* < 0.05; \*\**p* < 0.01; \*\*\**p* < 0.001. All scale bars indicate 250 μm.

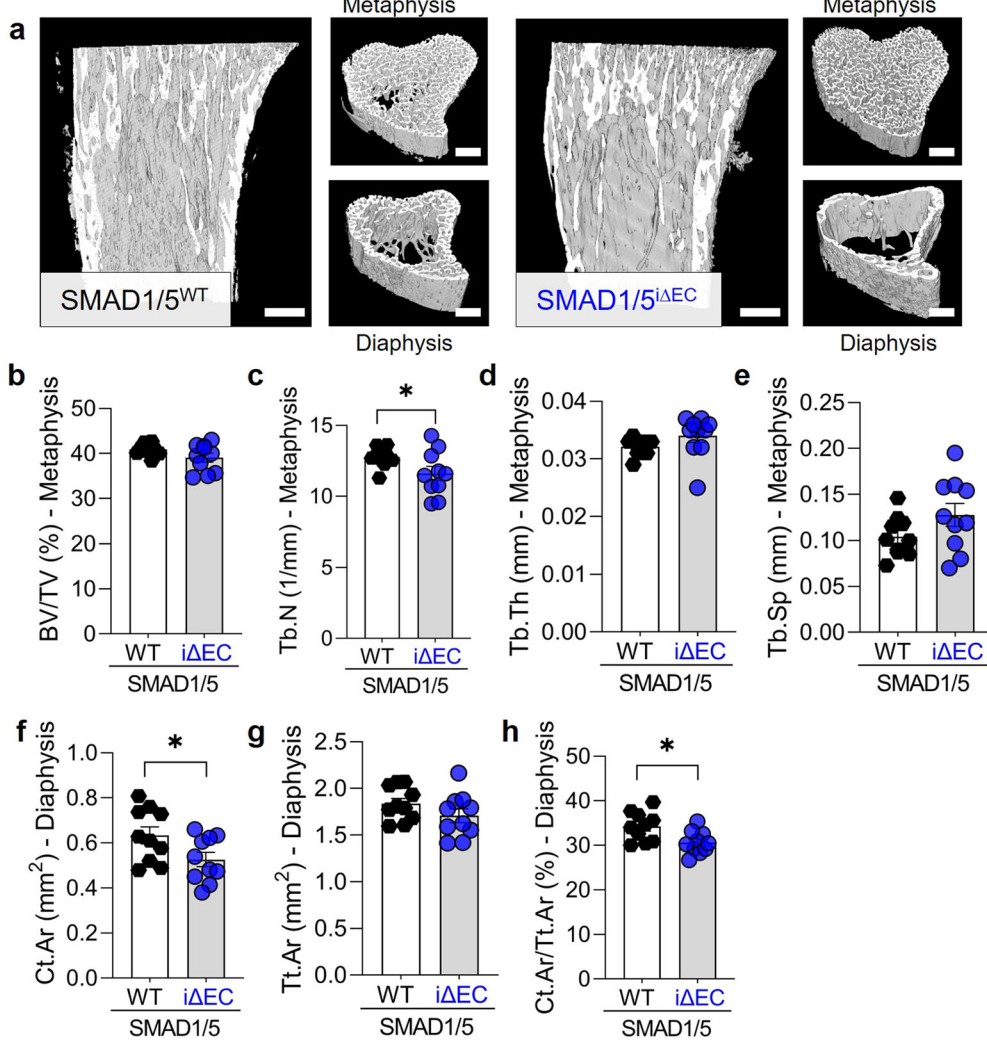

**Fig. 2 | Endothelial SMAD1/5 depletion after weaning decreased diaphyseal bone formation. a** μCT-based 3D rendering of the proximal tibia at P28. Quantitative μCT-based structural analysis (P28; $n^{WT}$ = 9; $n^{i\Delta EC}$ = 10) of **b** bone volume fraction (BV/TV), **c** trabecular number (Tb.N), **d** trabecular thickness (Tb.Th) and **e** trabecular separation (Tb.Sp) in the metaphysis or **f** cortical bone area (Ct.Ar),

**g** total cross-sectional area (Tt.Ar) and **h** cortical area fraction (Ct.Ar/Tt.Ar) in the diaphysis. Bar graphs show mean ± SEM and individual data points. Two-sample *t*-test was used to determine the statistical significance; *p*-values are indicated with \**p* < 0.05. All scale bars indicate 250 μm.

significantly increased the relative vessel volume (VV/TV) in both the metaphysis and diaphysis (*metaphysis p* = 0.037; diaphysis *p* < 0.001; Fig. 1c, e) and increased the relative vessel surface in the diaphysis (*p* = 0.001) (Fig. 1d, f). Measurement of vascular linear density (V.Li.Dn) indicated that SMAD1/5 depletion did not alter vessel number in the metaphysis (Fig. 1g), but significantly elevated vessel number in the diaphysis (*p* = 0.022; Fig. 1k). SMAD1/5 depletion significantly elevated the mean vessel width in both the metaphysis and diaphysis by 46.4% and 55.6%, respectively (*p* < 0.001), reducing the frequency of smaller capillaries (<0.04 mm) and increasing the frequency of larger vessels (Fig. 1h, i, l, m). SMAD1/5 depletion did not significantly alter vascular separation (i.e., spacing between vessels) in the metaphysis, but reduced vascular separation in the diaphysis (*p* < 0.001; Fig. 1j, n). These data demonstrate a critical role of postnatal endothelial SMAD1/5 signaling in shaping and maintaining the 3D morphology of both metaphyseal and diaphyseal vessels.

## Endothelial SMAD1/5 activity directs cortical bone formation during long bone growth

The postnatal long bones are characterized by endochondral ossification at the growth plate and cortical bone maturation[24]. To determine the role of endothelial SMAD1/5 activity in trabecular and cortical bone formation, we

examined the bone morphometrical parameters of the metaphyseal and diaphyseal regions of the tibia using μCT (Fig. 2a). EC-specific SMAD1/5 depletion significantly reduced trabecular number (Tb.N; *p* = 0.047), but did not alter bone volume fraction (BV/TV), trabecular thickness (Tb.Th), or separation (Tb.Sp) (Fig. 2b–e). SMAD1/5 depletion significantly reduced cortical bone area (Ct. Ar) and the cortical area fraction (Ct.Ar/Tt.Ar) in the diaphysis (*p* = 0.042 and *p* = 0.013, respectively; Fig. 2f–h). These findings indicate that endothelial SMAD1/5 signaling directs cortical bone morphogenesis in juvenile long bones.

## Angiogenic-osteogenic coupling in the metaphysis requires endothelial SMAD1/5 activity

Metaphyseal vessels couple angiogenesis and bone formation during endochondral ossification. These specialized capillaries exhibit a columnar structure, terminate at the growth plate in anastomotic arches, and associate with Osterix-expressing (OSX$^+$) osteoprogenitor cells[7]. To determine the role of postnatal SMAD1/5 signaling in metaphyseal vessel morphogenesis and angiogenic-osteogenic coupling, we first examined the number of vessel arches adjacent to the growth plate and the area of Endomucin (EMCN) and CD31-expressing vessels in the metaphysis (Supplementary Fig. 2). Postnatally, EMCN is a specific vascular marker for sinusoidal and venous

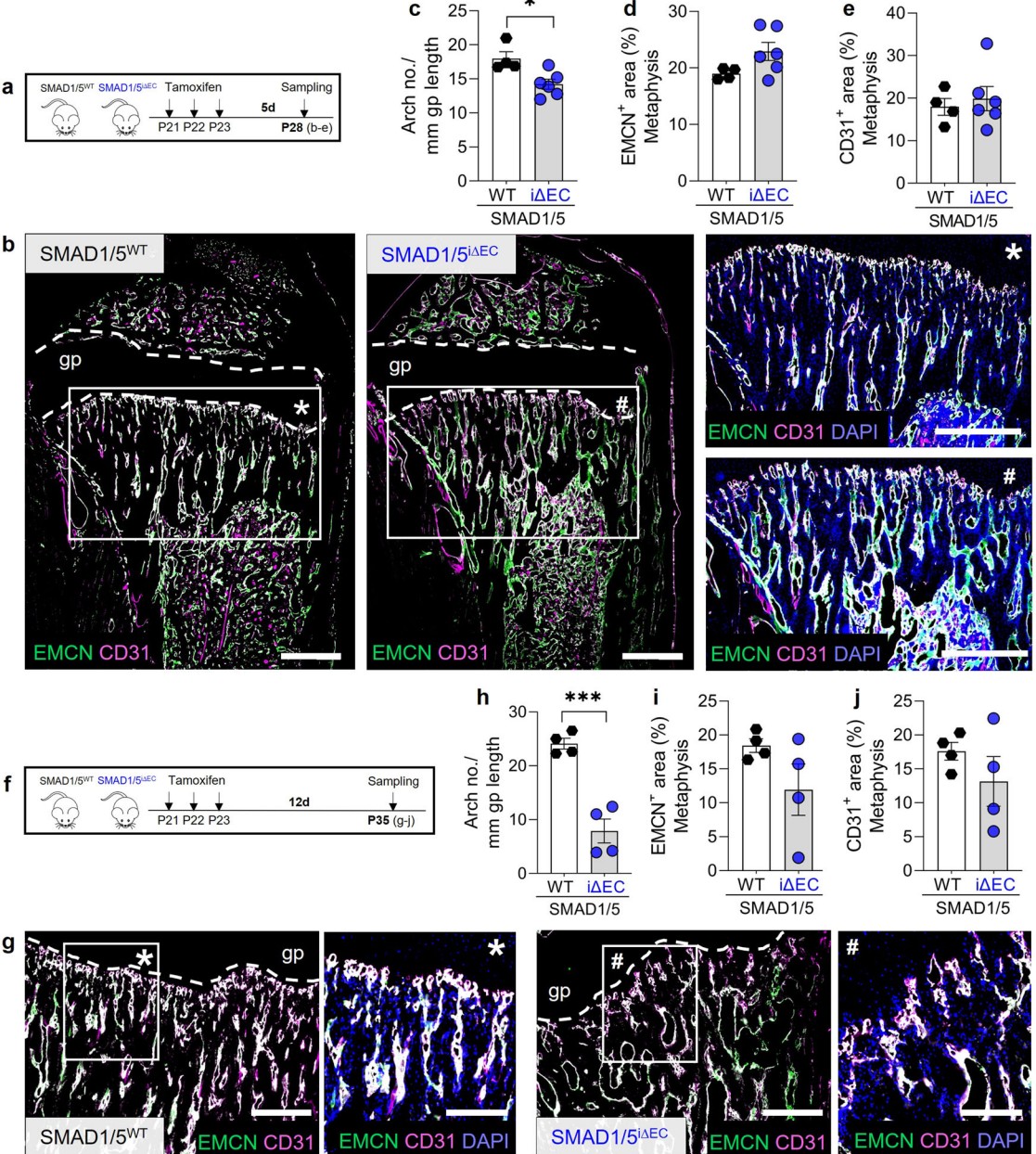

**Fig. 3 | Endothelial SMAD1/5 activity regulates to metaphyseal vessel morphology in juvenile mice. a** Tamoxifen treatment and short-term sampling scheme. Mice were injected postnatal day 19–21 (P19–21) and samples were collected at P28. **b** Representative images of EMCN and CD31 staining in the tibial metaphyseal and diaphyseal area showing EMCN, CD31 and DAPI staining in the metaphyseal area (P28; *$n^{WT}$ = 4; #$n^{i\Delta EC}$ = 6). Quantification of **c** arch number, **d** EMCN+ and **e** CD31+ areas (P28; $n^{WT}$ = 4; $n^{i\Delta EC}$ = 6). **f** Tamoxifen treatment and long-term sampling scheme. Mice were injected postnatal day 19–21 (P19–21) and samples were collected at P35. **g** Representative images of EMCN and CD31 staining in the tibial metaphyseal area and magnifications showing EMCN, CD31 and DAPI staining (P35; *$n^{WT}$ = 4; #$n^{i\Delta EC}$ = 4). Quantification of **h** arch number, **i** EMCN+ and **j** CD31+ areas (P35; $n^{WT}$ = 4; $n^{i\Delta EC}$ = 4). gp growth plate. Bar graphs show mean ± SEM and individual data points. Two-sample *t*-test was used to determine the statistical significance; *p*-values are indicated with *$p < 0.05$. All scale bars indicate **b** 500 µm, **g** 250 µm and 125 µm (magnifications).

vessels, while CD31 is additionally expressed by arteries, endothelial progenitors and other myeloid-lineage cells[7]. For the experimental design, we chose both short-term (7 days) and long-term (14 days) SMAD1/5 depletion from the endothelium of juvenile mice. Thus, we investigated cellular changes at P28 (7d after first tamoxifen injection; Fig. 3a) and P35 (14 days after first tamoxifen injection; Fig. 3f). EC-specific SMAD1/5 depletion resulted in aberrant vascular architecture (Fig. 3b) and reduced the number of anastomotic arches (mean difference = 3.8 ± 1.2 arches/mm; *p* = 0.02) at the chondro-osseus junction (Fig. 3c) but did not significantly alter EMCN+ and CD31+ areas in the metaphysis (Fig. 3d, e). By two-weeks post tamoxifen, EC-specific SMAD1/5 depletion resulted in an aberrant

columnar structure of the metaphyseal vessels and significant reduced anastomotic arch numbers (*p* < 0.001; Fig. 3g, h). Differences in EMCN+ and CD31+ areas in the metaphysis were not statistically significantly, but reflect a qualitative reduction (Fig. 3i, j). Together, these data demonstrate a role of endothelial SMAD1/5 signaling in short- and long-term morphogenesis of metaphyseal vessels.

Metaphyseal vessels physically associate with OSX+ osteoprogenitor cells and couple angiogenesis to osteogenesis during postnatal bone growth[7]. Therefore, we next evaluated the effects of endothelial SMAD1/5 depletion on OSX+ osteoprogenitors dynamics in the metaphysis by quantifying OSX+ cells at 7d and 14d post-tamoxifen injection (P28 vs. P35, respectively;

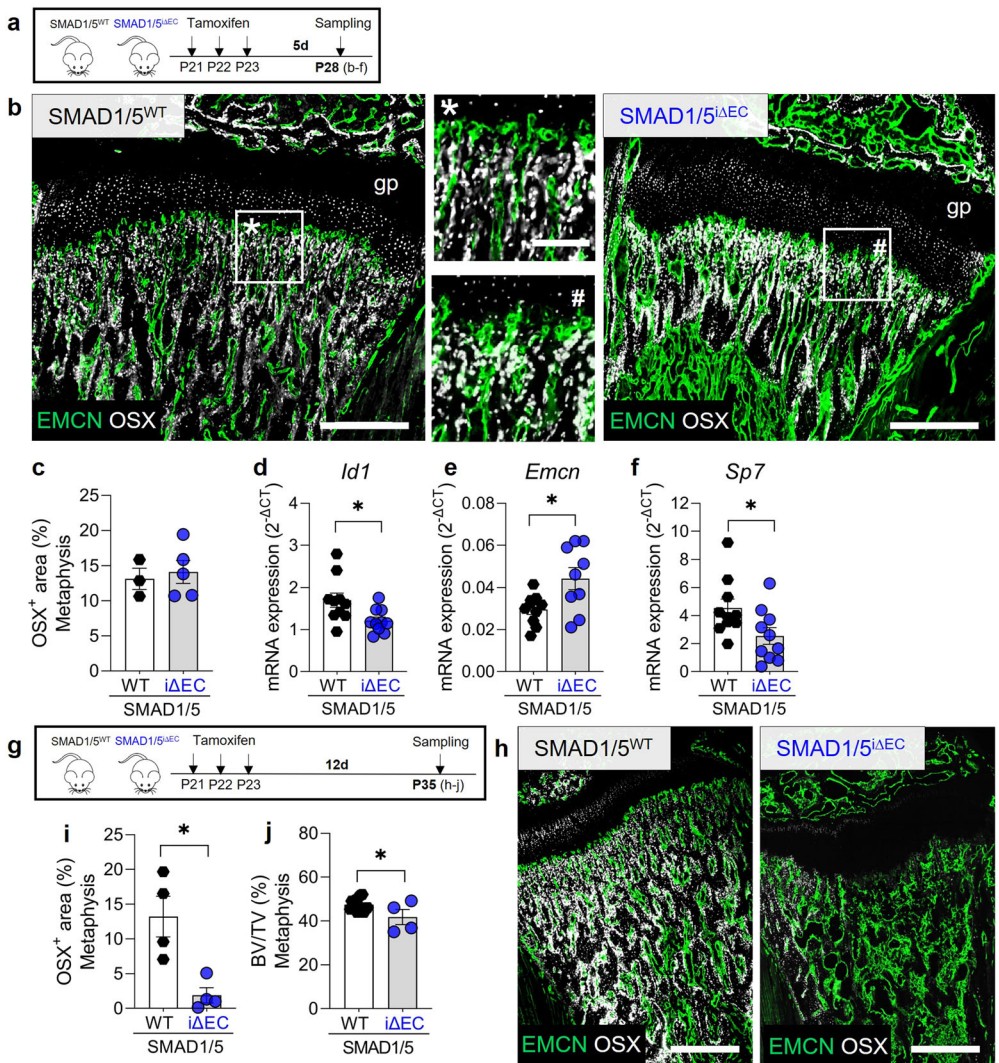

**Fig. 4 | Endothelial SMAD1/5 couples vessels to osteoprogenitors in metaphyseal juvenile bone. a** Tamoxifen treatment and short-term sampling scheme. Mice were injected postnatal day 19–21 (P19–21) and samples were collected at P28.
**b** Representative images of EMCN and OSX staining in the metaphyseal area (P28; $n^{WT} = 4$; $n^{i\Delta EC} = 6$). gp growth plate. Quantification of **c** OSX$^+$ cell area (P28; $n^{WT} = 4$; $n^{i\Delta EC} = 6$). Relative mRNA expression of **d** *Id1*, **e** *Emcn* and **f** *Sp7* normalized to *Hprt* (housekeeping gene) in the epi-/metaphysis (P28; $n = 10$). **g** Tamoxifen treatment and long-term sampling scheme. Mice were injected postnatal day 19–21 (P19–21)

and samples were collected at P35. **h** Representative images of EMCN and OSX staining in the metaphysis (P35; $n^{WT} = 4$; $n^{i\Delta EC} = 4$). Quantification of **i** OSX$^+$ cell area (P35; $n^{WT} = 4$; $n^{i\Delta EC} = 4$) and **j** BV/TV - quantitative µCT analysis (P35; $n^{WT} = 16$; $n^{i\Delta EC} = 4$). gp growth plate. Bar graphs show mean ± SEM and individual data points. Two-sample *t*-test or Mann Whitney U test (*Id1* RNA expression) was used to determine the statistical significance; *p*-values are indicated with *$p < 0.05$. All scale bars indicate 500 µm (**b**, **h**) or 125 µm (magnifications **b**).

Fig. 4a, g). EC-specific depletion of SMAD1/5 did not significantly alter OSX$^+$ cells in the metaphysis at 7 days post-depletion (P28; Fig. 4b, c; Supplementary Fig. 3). Bulk gene expression analysis was performed on metaphyseal and epiphyseal tissue to evaluate expression of the canonical SMAD1/5-target gene, *Id1*[25]. As expected, *Id1* expression was significantly lower in the meta-/epiphysis of SMAD1/5$^{i\Delta EC}$ mice ($p = 0.03$; Fig. 4d). Although EMCN$^+$ area was not significantly elevated (Fig. 3d), EC-specific depletion of SMAD1/5 increased *Emcn* mRNA abundance (fold change between means 1.5; $p = 0.02$; Fig. 4e). *Sp7* (OSX) mRNA was also significantly reduced (44% lower, $p = 0.04$) by endothelial SMAD1/5 deactivation at P28 (Fig. 4f). By 2 weeks post tamoxifen (Fig. 4g), EC-specific SMAD1/5 depletion significantly and markedly decreased the abundance of OSX$^+$ osteoprogenitors ($p = 0.011$) in the metaphysis (Fig. 4h, i). Consistently, quantitative µCT analysis revealed significantly reduced metaphyseal bone volume fraction (BV/TV) upon EC-specific SMAD1/5 depletion ($p = 0.014$; Fig. 4j). Together, these data indicate that endothelial SMAD1/5 activity regulates metaphyseal vessel morphogenesis and is

required for maintenance of osteoprogenitor cells in the metaphysis, functionally coupling angiogenesis and osteogenesis in juvenile bone.

## Loss of metaphyseal vessel integrity results in accumulation of hypertrophic chondrocytes in the growth plate

The metaphyseal capillaries function not only to support network connectivity and osteoprogenitor mobilization, but together with surrounding cells[26], actively degrade the hypertrophic cartilage to enable endochondral ossification[26]. Therefore, we next asked whether the disruption of the metaphyseal vessel structures caused by endothelial SMAD1/5 depletion affected the morphogenesis and remodeling of the hypertrophic cartilage at the chondro-osseous junction. For the experimental design, we chose the same procedure as for the osteoprogenitor analysis to address growth plate remodeling dynamics. Thus, we investigated growth plate changes at P28 (7d after first tamoxifen injection; Fig. 5a) and also P35 (14 days after first tamoxifen injection; Fig. 5e). Depletion of SMAD1/5 activity in ECs did not significantly alter cell morphology, thickness, or hypertrophic chondrocyte fraction in the

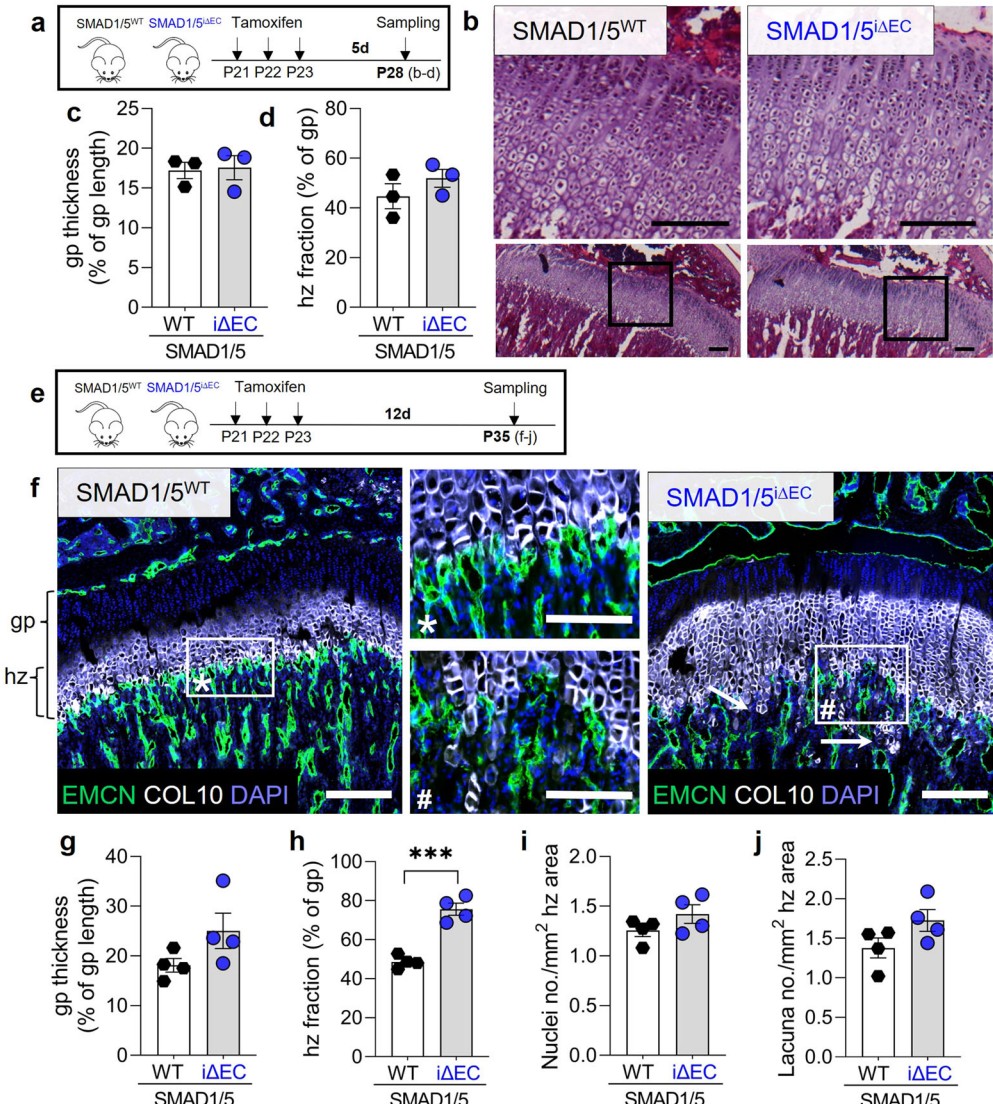

**Fig. 5 | Endothelial SMAD1/5 signaling regulates hypertrophic chondrocyte remodeling at the chondro-osseous junction in juvenile mice. a** Tamoxifen treatment and short-term sampling scheme. Mice were injected postnatal day 19–21 (P19–21) and samples were collected at P28. **b** Representative images of H&E staining at P28. Quantification of **c** growth plate thickness relative to the growth plate length and **d** hypertrophic zone fraction at P28 ($n^{WT}$= 3; $n^{i\Delta EC}$ = 3). **e** Tamoxifen treatment and long-term sampling scheme. Mice were injected postnatal day 19–21 (P19–21) and samples were collected at P35. **f** Representative images of EMCN, COL10 and DAPI staining in the epi-/metaphysis at P35. gp growth plate, hz hypertrophic zone; arrows indicate penetration of COL10 positive chondrocyte columns into the metaphyseal vascular area. Quantification of **g** growth plate thickness relative to the growth plate length, **h** hypertrophic zone fraction, **i** nuclei as well as **j** lacuna number in the hypertrophic zone area ($n^{WT}$ = 4; $n^{i\Delta EC}$ = 4). Bar graphs show mean ± SEM and individual data points. Two-sample $t$-test was used to determine the statistical significance; $p$-values are indicated with ***$p < 0.001$. All scale bars indicate 250 μm **b**, **f** or 125 μm (magnifications **f**).

growth plate at 7 days post-tamoxifen (Fig. 5b–d). Consistently, SMAD1/5 depletion did not alter metaphyseal mRNA expression of *Mmp9, Ctsk, Adamts1* and *Timp1* at 7 days post-tamoxifen (Supplementary Fig. 4). However, by P35, 14 days post-tamoxifen, EC-specific SMAD1/5 depletion resulted in dysmorphogenesis of the anastomotic arches at the chondro-osseous junction (Fig. 5f; arrows; cf. Fig. 3h, g) and a significant enlargement of the hypertrophic zone (hz) of the growth plate (Fig. 5g, h). Quantification of the total growth plate size indicated that reduced endothelial SMAD1/5 activity did not induce a general enlargement of the total growth plate ($p = 0.1$; Fig. 5g) but a shift of zonal distribution with a significant increase in the relative COL10$^+$ area (27% increase; $p = 0.0003$; Fig. 5h). Chondrocytes occupy lacunae in the extracellular matrix which can be counted in parallel to DAPI$^+$ nuclei for assessment of growth plate cellularity. The number of DAPI$^+$ cells and chondrocyte lacuna in the COL10$^+$ area was slightly increased, suggesting an increase in cellular quantity rather than a volumetric enlargement (Fig. 5i, j). It has been shown that MMP9 activity in metaphyseal vessels directs growth

plate size during bone development[26]. Therefore, we performed additional MMP9 staining in the metaphysis and found evidence for reduced metaphyseal MMP9 abundance by P35, 14 days post-tamoxifen (Supplementary Fig. 5). Together, these data establish the necessity of ongoing SMAD1/5 signaling in maintenance of metaphyseal vessel-mediated resorption of hypertrophic cartilage and growth plate remodeling.

### Endothelial SMAD1/5 signaling regulates vascular maturation in diaphyseal sinusoidal capillaries

Diaphyseal, bone marrow-associated vessels have sinusoidal structure, and functionally couple with hematopoiesis in the bone marrow[7,27]. To determine the role of postnatal SMAD1/5 signaling in diaphyseal vessel morphogenesis and maintenance, we examined the number of vascular loops and the area of EMCN and CD31-expressing vessels at P28 and P35 (Fig. 6a, g) in the diaphysis[7] (Supplementary Fig. 2). Endothelial SMAD1/5 depletion significantly increased the number and size of diaphyseal vascular

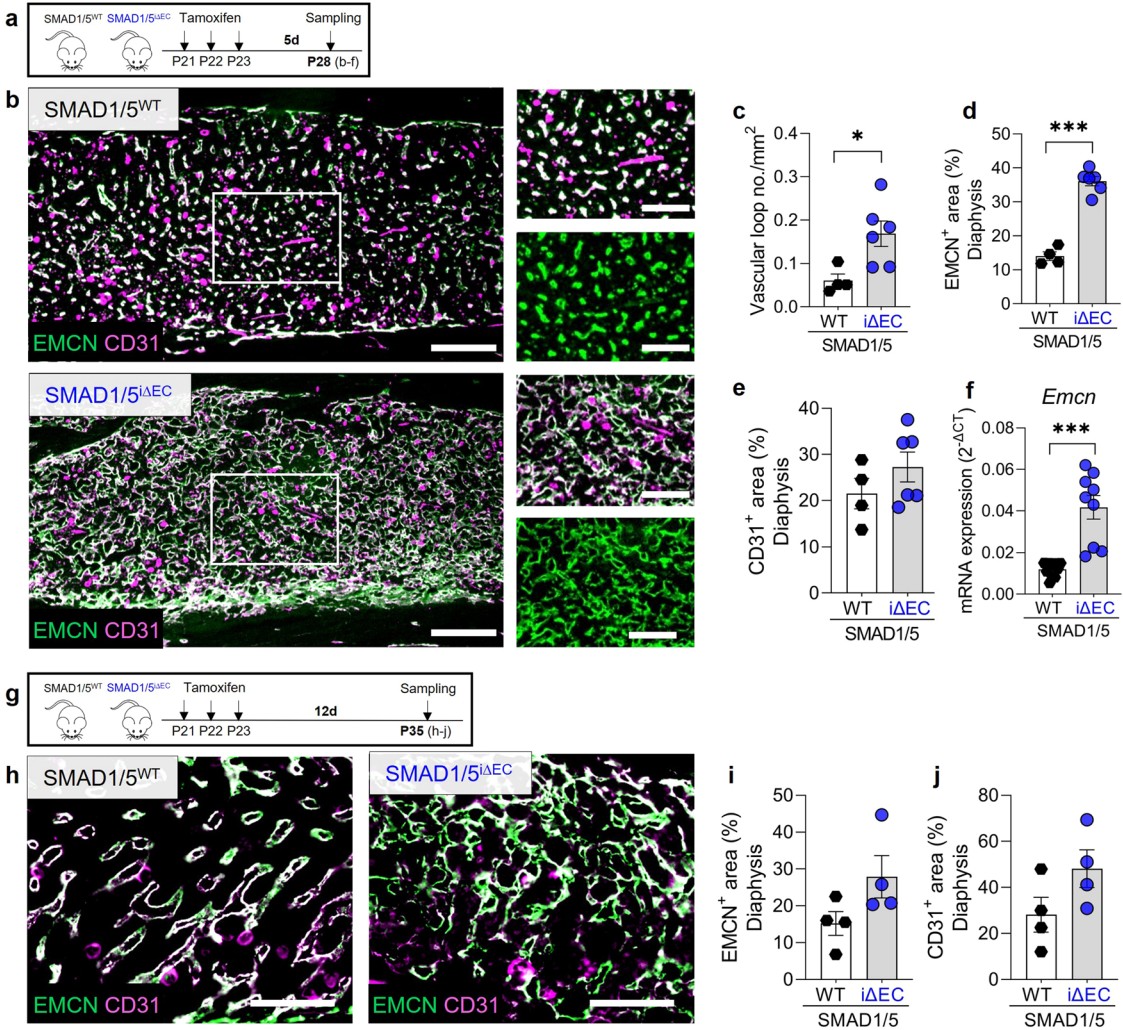

**Fig. 6 | Endothelial SMAD1/5 promotes maturation and maintenance of diaphyseal sinusoidal capillaries. a** Tamoxifen treatment and short-term sampling scheme. Mice were injected postnatal day 19–21 (P19–21) and samples were collected at P28. **b** Representative images of EMCN and CD31 staining in the diaphysis (P28; $n^{WT} = 4$; $n^{i\Delta EC} = 6$). Quantification of **c** number of vascular loops per mm², **d** relative EMCN⁺ and **e** CD31⁺ area (P28; $n^{WT} = 4$; $n^{i\Delta EC} = 6$). **f** Relative mRNA expression analysis of *Emcn* normalized to *Hprt* (housekeeping gene) in the diaphysis (P28; $n = 10$). **g** Tamoxifen treatment and long-term sampling scheme. Mice were injected postnatal day 19–21 (P19–21) and samples were collected at P35. **h** Representative images of EMCN and CD31 staining in the diaphysis (P35; $n = 4$). Quantification of **i** relative EMCN⁺ and **j** CD31⁺ area (P35; $n = 4$). Bar graphs show mean ± SEM and individual data points. Two-sample *t*-test was used to determine the statistical significance; *p*-values are indicated with \**p* < 0.05; \*\*\**p* < 0.001. All scale bars indicate 250 μm (**b**) or 125 μm (magnifications **b**, **h**).

loops ($p = 0.02$; Fig. 6b, c) at 7 days post-tamoxifen, underlining the hypervascularity indicated in the CECT data (Fig. 1). Endothelial-conditional SMAD1/5 depletion increased the EMCN⁺ area (mean difference = 22% ± 2%; $p < 0.001$; Fig. 6d) but differences in CD31⁺ area were not significant (mean difference = 5.8% ± 4.8%; $p = 0.27$; Fig. 6e). Consistently, *Emcn* mRNA was elevated 4-fold in the diaphyseal bone marrow area in SMAD1/5^iΔEC mice ($p < 0.001$; Fig. 6f). By P35, 14 days post-tamoxifen (Fig. 6g), EC-specific SMAD1/5 depletion resulted in more pronounced dysmorphogenesis of the diaphyseal vessels, which did not allow for vascular loop quantification due to loss of network integrity (Fig. 6h). EMCN⁺ area (Fig. 6i) and CD31⁺ area (Fig. 6j) were increased although not significantly. As clinical vascular disorders caused by genetic defects in BMP-ALK1 signaling (hereditary hemorrhagic telangiectasia; HHT) are also characterized by vessel wall fragility, we stained for Ter119⁺ erythrocytes to assess extravascular red blood cell abundance. Because erythropoiesis does occur in the bone marrow, we additionally evaluated CD71⁺ erythroid progenitors (Supplementary Fig. 6). We observed an increase in extravascular red blood cells, consistent with a HHT-related vascular permeability phenotype. However, we also observed qualitatively decreased CD71 positivity at later stages (P35; Supplementary Fig. 6). These observations

suggest a potential role of vascular SMAD1/5 activity in vascular permeability and barrier function but may also indicate regulation of erythropoiesis. Together, these data show that endothelial SMAD1/5 activity is essential to maintain the diaphyseal sinusoidal capillary phenotype, with SMAD1/5 depletion inducing excessive formation of large vascular loops.

**Endothelial SMAD1/5 activity is required for metaphyseal and diaphyseal maintenance during early adolescence**

Since the bone marrow vasculature undergoes continuous remodeling during postnatal and adolescent development, we next investigated the effects of EC-specific depletion of SMAD1/5 in more mature mice. Mice were injected with tamoxifen at P42 and samples were collected 7 or 14 days later (P49 and P56, i.e., 4 and 5 weeks post-weaning, respectively; Fig. 7a; Supplementary Fig. 7). Analysis of metaphyseal vessels at 7 days after endothelial-conditional SMAD1/5 depletion (P49) revealed impaired columnar structure (Fig. 7b). Moreover, as in younger mice, 7 days of SMAD1/5 depletion reduced the number of anastomotic arches ($p = 0.02$) adjacent to the growth plate (Fig. 7c) and did not significantly alter EMCN⁺ area but significantly reduced CD31⁺ area ($p = 0.035$; Fig. 7d, e). Analysis of diaphyseal vessels at 7 days after endothelial-conditional SMAD1/5

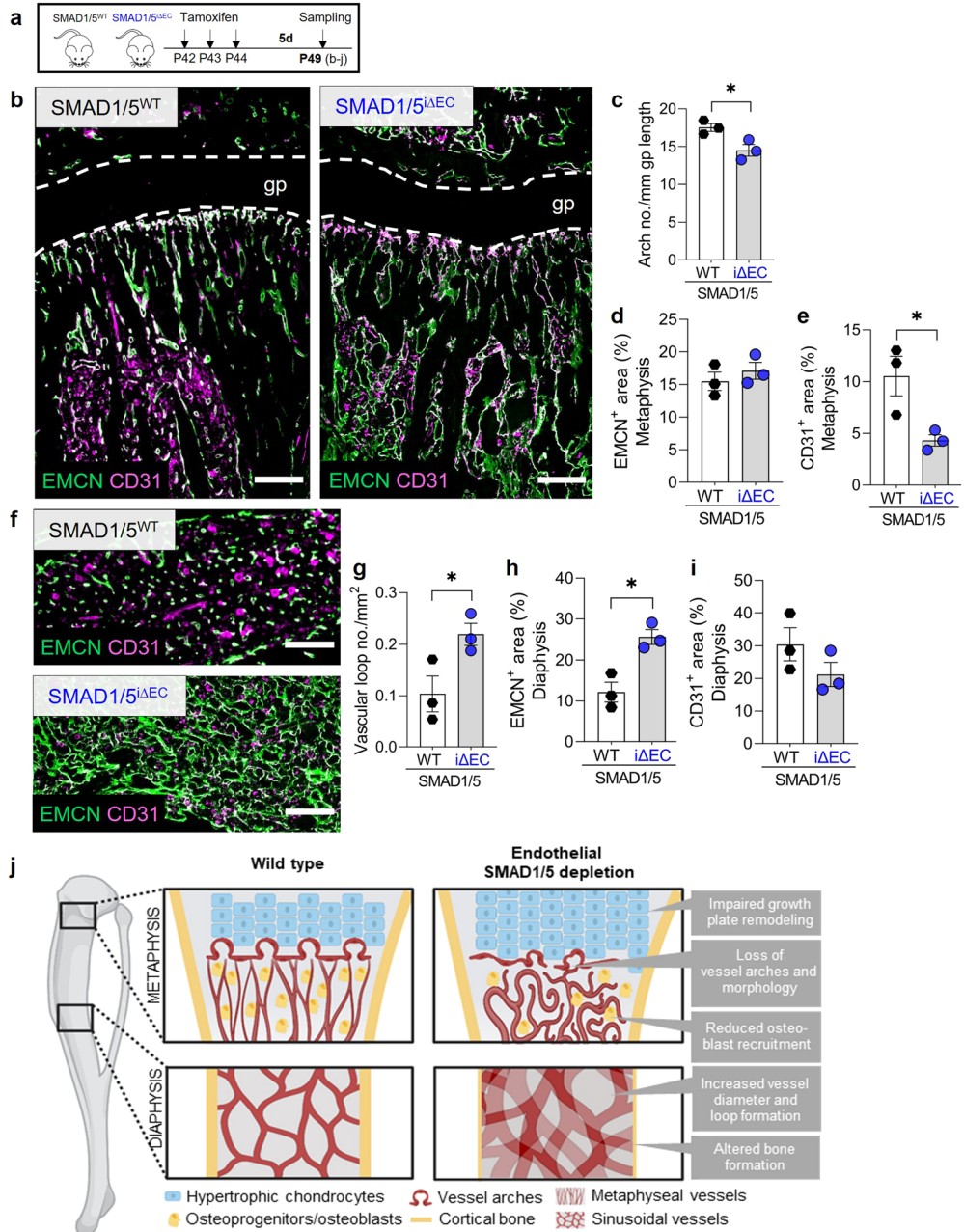

**Fig. 7 | Endothelial SMAD1/5 maintains morphology and function of metaphyseal and diaphyseal capillaries during early adolescent. a** Tamoxifen treatment and short-term sampling scheme. Mice were injected postnatal day 42–44 (P42–44) and samples were collected at P49 (7–4 weeks post-weaning). **b** Representative images of EMCN and CD31 staining in the metaphysis (P42; $n^{WT}$ = 3; $n^{i\Delta EC}$ = 3). Quantification of **c** arch number, **d** relative EMCN$^+$ and **e** CD31$^+$ area. **f** Representative images of EMCN and CD31 staining in the diaphysis (P49; $n^{WT}$ = 3;

$n^{i\Delta EC}$ = 3). Quantification of **g** number of vascular loops per mm², **h** relative EMCN$^+$ and **i** CD31$^+$ area. Bar graphs show mean ± SEM and individual data points. Two-sample *t*-test was used to determine the statistical significance; *p*-values are indicated with **p* < 0.05. All scale bars indicate 250 μm (**b**, **f**). **j** Graphical summary on effects of endothelial SMAD1/5 depletion on metaphyseal and diaphyseal vessel and bone formation during juvenile and early adolescent long bone growth. Illustration has been created with BioRender.com.

depletion (P49) revealed significantly increased vascular loop formation (*p* = 0.047) (Fig. 7g), as in younger mice and increased EMCN$^+$ area (*p* = 0.011) with no differences in CD31$^+$ area (*p* = 0.21; Fig. 7h, i). These alterations to the metaphyseal and diaphyseal vasculature were pronounced at P56 (14 days post-tamoxifen injection) (Supplementary Fig. 7). Moreover, OSX staining in the P56 metaphysis indicated qualitatively reduced osteoprogenitor abundance at 14 days post-depletion, but this reduction was less dramatic than in young mice at P35 (Supplementary Fig. 7, cf. Figure 4i). Similarly, endothelial SMAD1/5 depletion disrupted vascular loop formation at the chondro-osseous junction and disorganized the cartilage septum, but without enlargement of the hypertrophic zone observed

in juvenile mice (Supplementary Fig. 7, cf. Fig. 5f). Together, these data support a model in which endothelial SMAD1/5 activity regulates metaphyseal vascular sprouting dynamics, maintains diaphyseal vascular stability, and coordinates growth plate remodeling and osteoprogenitor recruitment dynamics.

## Discussion

Here, we show that endothelial SMAD1/5 activity sustains skeletal vascular morphogenesis and function and coordinates growth plate remodeling and osteoprogenitor recruitment dynamics during juvenile and adolescent bone growth (Fig. 7j). We found that endothelial cell-conditional SMAD1/5

depletion in juvenile mice caused hypervascularity in both metaphyseal and diaphyseal vascular compartments, resulting in altered trabecular and cortical bone formation. Short-term and long-term SMAD1/5 depletion, in both juvenile and adolescent mice, induced excessive sprouting and disrupted the morphological structure of metaphyseal vessels, and impaired anastomotic loop formation at the chondro-osseous junction. SMAD1/5 depletion progressively arrested osteoprogenitor recruitment to the primary spongiosa and, in the long-term, impaired growth plate resorption. Finally, in the diaphyseal sinusoids, endothelial SMAD1/5 activity was necessary to maintain vessel architecture, with SMAD1/5 depletion inducing excessive formation of large vascular loops, and potential hyperpermeability. Together, these data show that SMAD1/5 signaling in the endothelium preserves skeletal vessel structure and function and couples angiogenesis to osteogenesis in juvenile and adolescent bone.

Juvenile skeletal growth requires dynamic adaptation of bone formation accompanied by a substantial adjustment of the bone vasculature. Specialization of vascular morphology within the long bones initiates at postnatal day 6, with metaphyseal capillaries acquiring a column-like structure and diaphyseal capillaries forming a sinusoidal network[28]. Murine long bone growth evolves rapidly until P14, reaching a steady growth phase between P14 and P42[29,30]. This is in accordance with the already described rapid decline of the metaphyseal vessels over the first 4 weeks postnatally[7]. We found that EC-specific SMAD1/5 depletion at P21 resulted in a substantial enlargement of the diaphyseal vasculature with less dramatic changes in the metaphyseal vasculature at P28. This observation suggests that endothelial SMAD1/5 signaling (i) directs morphogenesis of both metaphyseal and diaphyseal vessels in juvenile bone and (ii) maintains vascular stability in the diaphysis between P14 and P28. These changes further contributed to altered bone formation. Cortical bone was also impaired after short-term depletion of endothelial SMAD1/5 activity (P28). Although our data do not reveal the mechanisms which induce the changes in the cortical bone, we speculate that it might be a result of the changes in the metaphyseal bone or due to an impairment in transcortical vessels, as the process of corticalization is impacted by metaphyseal bone formation and subsequent remodeling[31,32]. In addition, cortical bone has been shown to have transcortical blood vessels which may impact cortical bone morphogenesis, though this has not yet been examined[33]. Further studies are needed to define the mechanisms of angiogenic-osteogenic coupling in the cortical bone during development and remodeling.

At the chondro-osseous junction, Osterix-expressing osteoprogenitors spatially localize with the metaphyseal endothelium and couple osteogenesis to angiogenesis via multiple pathways, including Notch signaling[7,34]. Comparable to the shape-maintaining function of endothelial Notch signaling[34], here we show that SMAD1/5 activity in metaphyseal endothelium is crucial to maintain their archetypal columnar structure and new arch formation in both juvenile (P21) and adolescent (P42) bones. DLL4-Notch signaling is responsible for tip and stalk cell competence in the metaphysis and is driven by crosstalk between ECs and chondrocytes (via VEGF, Noggin)[34]. We demonstrated previously that during mouse embryo development, Notch and SMAD1/5 signaling synergize to balance selection of tip and stalk cells in retinal vascular sprouting[21]. Synthesizing these findings with our present results, we posit that the alterations in metaphyseal vessel angiogenesis result from the disrupted Notch/SMAD1/5 synergy in the bulging vessels. In addition, BMP2/6/7, which signal through SMAD1/5, are abundant in bone[12,14]. These ligands guide endothelial tip cell competence via type I receptors (ALK2, ALK3, ALK6), in conjunction with BMP type II receptor[35], suggesting that bulging angiogenesis by vessels in the metaphysis may be regulated by BMP-SMAD signaling. Consistently, we observed profound disruption of angiogenic-osteogenic coupling in the metaphysis, with reduced *Sp7* mRNA expression at 7 days post-depletion and near complete abrogation of OSX-expressing cells in the metaphysis after 14 days in juvenile mice. Since OSX⁺ cells substantially expand during the first 4 weeks postnatally in the metaphysis[28], these time-dependent findings indicate the requirement of continued endothelial SMAD1/5 activity in osteoprogenitor survival and recruitment during endochondral bone

growth. Further studies are required to investigate the angiogenic-osteogenic crosstalk mechanisms and the fate of the osteoprogenitors upon endothelial SMAD1/5 depletion.

Endochondral bone formation at the chondro-osseous junction requires neovascular invasion and growth plate remodeling. Previous studies reporting enlargement of the growth plate, especially the hypertrophic zone, upon disruption of the growth plate-adjacent vasculature by inhibition of VEGF signaling[1,36] or endothelial MMP9 depletion[26]. Consistent with these data, we found that EC-specific SMAD1/5 depletion resulted in a significant enlargement of the hypertrophic zone of the growth plate. Therefore, dysmorphogenesis of the metaphyseal vessels, altered tip and stalk cell formation, and reduced expression of MMP9 at the chondro-osseous junction could explain the enlargement of the hypertrophic zone of the growth plate by EC-specific SMAD1/5 depletion. This is further supported by the observation in retinal angiogenesis that BMP4-SMAD1/5 signaling regulates endothelial MMP9 function[37]. Based on our finding that the number of DAPI⁺ cells and chondrocyte lacunae in the COL10⁺ area were only slightly increased, we propose that EC-specific SMAD1/5 inactivity affected the removal of cartilage matrix and the transition from hypertrophic chondrocytes to bone rather than chondrocyte hypertrophy, per se[38].

While growth plate remodeling and endochondral ossification are mediated by metaphyseal vessels, the long bone diaphysis is populated by sinusoidal vessels, which are maintained in a homeostatic state with relatively slower physiological remodeling[7]. Kusumbe et al. suggest that diaphyseal vessels emerge through maturation of metaphyseal capillaries[7,28]. While there is evidence suggesting constant remodeling and volume adaptations in the diaphyseal capillary network[39], the dynamics and underlying mechanisms are mostly unknown. We previously showed that EC-specific depletion of SMAD1/5 during early postnatal retinal angiogenesis resulted in arteriovenous malformations, a reduced number of tip cells, and hyperdensity in the retinal vascular plexus[22]. These findings mirror the diaphyseal vessel changes, characterized by significant hyper-density and aberrant vascular loop formation. We observed progressive emergence of diaphyseal vessels with significantly elevated EMCN and CD31 expression upon EC-specific depletion of SMAD1/5. Tip and stalk cell selection during sprouting angiogenesis is guided by DLL4/Notch interaction, with tip cells showing higher expression of DLL4[34]. Previously, we found that endothelial SMAD1/5 specifically regulates Notch-mediated tip cell formation in the E9.5 mouse hindbrain[21]. Thus, the hyper-dilatation of the diaphyseal vasculature may be a result of pronounced bulging angiogenesis (sprouting) in the diaphyseal vessels and progressive conversion to a metaphyseal-like phenotype, including an increase in tip-like endothelial cells, upon cessation of SMAD1/5 signaling. Vascular homeostasis, quiescence, and maturation are controlled by BMP9/10 signaling via ALK1-BMPR2 complexes activating SMAD1/5[40]. BMP9/10-ALK1-SMAD1/5 signaling may therefore modulate homeostatic signaling in type L vessel maturation and phenotypic maintenance. Together, these findings suggest a central role of endothelial SMAD1/5 in maintenance of sinusoidal vascular homeostasis.

A comparable phenotype of hyper-dilated and functionally leaky vessels has been described in mouse embryos with a global loss of the BMP receptor Activin receptor-like kinase 1 (ALK1) or adult mice with an endothelial-specific ALK1 knockout[41,42]. Genetic defects in ALK1 signaling cause the autosomal dominant vascular disorder, hereditary hemorrhagic telangiectasia (HHT), which causes arteriovenous malformations (AVM) and vessel wall fragility, resulting in a risk for fatal hemorrhage in human patients[43]. Arteriovenous malformations in human HHT bone marrow have been described[44].

In conclusion, this study identifies SMAD1/5 signaling in endothelial cells as an essential regulator of vascular formation, maturation, and homeostasis in juvenile and adolescent long bones, and as a mediator of angiogenic-osteogenic coupling. Our findings underline the importance of functional BMP-SMAD signaling in long bone vasculature and may inform clinical management of congenital diseases like HHT[44] and the development

of new therapies for enhancing vascularized bone repair and regeneration[15,45–47].

## Limitations

Based on our experimental design, we cannot draw conclusions on the functional role of SMAD1/5 in angiogenic-osteogenic coupling during embryogenesis and early postnatal bone development. Further studies will be necessary to dissect these early timepoints which exhibit more rapid cellular dynamics and unique cell populations compared to juvenile and adolescent skeletal formation. We have previously reported that a constitutive EC-specific depletion of SMAD1/5 activity is embryonically lethal[21,22], so continued study using the inducible system is warranted. In our study, we found that the serious malformations in the vascular system precluded analysis of samples collected at later timepoints after tamoxifen induction (14 days). This resulted in lower sample sizes in the analysis at P35 and P56. In addition, our experimental approach designed to detect differences according to sex as an independent variable, but both sexes were included in the study and equal distribution of data did not display evidence of sexual dimorphism.

## Methods

### Breeding strategy and housing

Mice were housed and bred in the Animal Facility at KU-Leuven (Belgium) and all animal procedures were approved by the Ethical Committee (P039/2017, M007, M008). Breeding was performed based on an already established scheme[22] described in the following. In detail, homozygous mice caring the Smad1/Smad5 floxed alleles (Smad1$^{fl/fl}$;Smad5$^{fl/fl}$) were paired with endothelium-specific tamoxifen-inducible Cre mice expressing (Cdh5-CreERT2$^{tg/0}$). Subsequently, dams (Smad1$^{fl/fl}$;Smad5$^{fl/fl}$) were crossed with the obtained Cdh5-CreERT2$^{tg/0}$;Smad1$^{fl/+}$;Smad5$^{fl/+}$ mice. The resulting Cdh5-CreERT2$^{tg/0}$;Smad1$^{fl/fl}$;Smad5$^{fl/fl}$ pups were injected intraperitoneally with tamoxifen (500 µg; Sigma Aldrich) at (i) postnatal day 19, 20 and 21 (P21) or (ii) postnatal day 42 (6 week old) to create EC-specific double knockout pups (SMAD1/5$^{iΔEC}$). Pups were killed at P28 or P35, or at P49 (7 week old) or P56 (8 week old). Mice have a mixed background of CD1 and C57BL/6. All experiments were conducted using Cre-negative littermate controls. Genotyping of recombined alleles was done after sample collection as previously described[21]. For breeding, mice were housed in pairs (one male and one female) in IVC Eurostandard Type II clear-transparent plastic cages (two animals per cage) covered with a wire lid and built-in u-shaped feed hopper and closed with a filter top in a SFP barrier facility. Weaning was performed at an age of ~3 weeks while littermates were housed together with 5 mice in Eurostandard Type II cages and transferred to a semi-barrier facility with IVC cages. As bedding material, fine wood chips and Nestlets for nesting were provided as well as plastic houses for environmental enrichment. The room temperature was constant in both facilities between 20 and 22 °C and a 12/12 h light/dark cycle with lights on at 700 h and off at 1900 h. Mice received standard diet and tap water *ad libitum*. Mice were killed by cervical dislocation. Male and female mice were used for investigations and sex-specific differences were not analyzed. All experiments and analyses were conducted with samples from at least 3 different litters/experiments. Samples were assigned random numbers to ensure unbiased analysis by the experimenter (genotype unknown; blinding). Group assignments were revealed upon complete analysis.

### Contrast-enhanced microfocus X-ray computed tomography

Right tibias from SMAD1/5$^{WT}$ and SMAD1/5$^{iΔEC}$ mice ($n = 10$ per group) were collected and fixed in 4% paraformaldehyde (PFA; Sigma Aldrich) in PBS overnight (12 h) at 4 °C. Samples were stored in PBS at 4 °C until further use to allow for consistent staining of all samples with an X-ray contrast-enhancing staining agent (CESA). The distal part of the tibia was cut to open the shaft and allow for uniform distribution of the CESA solution. Samples were stained for 1 week with a Hafnium-substituted Wells-Dawson polyoxometalate (POM) solution (35 mg/ml PBS) at 4 °C under constant shaking as established previously[48]. The Tungsten-containing POM compound therefore stains the hematopoietic tissue (based on electrostatic interactions) while blood vessels are empty with no staining which allows the subsequent visualization. High-resolution microfocus computed tomography (µCT) imaging was performed with a GE NanoTom M (GE Measurement & Control) at 60 kV and 140 µA, with a 0.2 mm filter of aluminum and a voxel size of 2 µm. The exposure time was 500 ms, and 2400 images were acquired over 360° using the fast scan mode, resulting in 20 min acquisition time. During reconstruction (Datos-x, GE Measurement & Control), a beam hardening correction of 7 and a Gaussian filter of 6 was used. Detailed structural analysis of all datasets was performed using CTAn (version 1.16) and DataViewer (both Bruker Corporation). Volumes of interest (VOIs) of 301 images (0.6 mm) were analyzed in the metaphysis and diaphysis, respectively. To determine the starting point of the metaphyseal and diaphyseal area, the image displaying the middle part of the growth plate was manually determined (GP). The start of the metaphyseal VOI was determined 300 images downstream of the GP, while the diaphysis started 100 images under the end of the metaphyseal area, representing the transition zone between meta- and diaphysis. Thresholding for binarization of the vessels was manually performed based on the histogram, while for bone binarization, automatic Otsu thresholding was applied. Manually drawn ROIs of the bone area (outer cortical surface) were specified with the ROI shrink-wrap tools stretching over holes with a diameter of 60 pixels and independent objects were removed using the despeckle tool. For analysis, the provided task set for 3D analysis was employed including analysis of structure separation distribution for the vessels. 1000 images of exemplary samples were used for 3D rendering and visualization (CTvox; version 3.2.0; Bruker Corporation). Relevant trabecular and cortical outcomes were selected and calculated following Bouxein et al.[49].

Additional samples (P35) were with a SkyScan 1172 high-resolution µCT (Bruker Corporation). Tibias were scanned with a voxel size of 8 µm, a source energy of 70 kV, 142 µA, a rotation step of 0.3 degree and an 0.5 mm aluminum filter. Reconstruction was performed using NRecon (Bruker Corporation), applying ring artifact reduction and beam hardening corrections. CTAn (version 1.20.3.0) was used for 3D analyses and the volume of interest was manually determined with the metaphyseal VOI reaching 300 images downstream of the GP.

### Classical 2D histology and immunofluorescence

For immunofluorescence, tibias were fixed in 4% PFA in PBS for 6–8 h at 4 °C. Samples were cryo-embedded (SCEM medium, Sectionlab) after treatment with an ascending sucrose solution (10, 20, 30%) for 24 h each. Sectioning was performed using a cryotape (Sectionlab) and sections were stored at -80 °C until further use. For immunofluorescence staining, sections were airdried for 10 min before being hydrated in PBS (5 min). Blocking solution contained 10% donkey or goat serum in PBS (30 min) and antibodies were diluted in PBS/0.1% Tween20/5% donkey or goat serum (Sigma Aldrich). The following primary antibodies and secondary antibodies were used (staining durations are individually provided): pSMAD1/5/9 (Cell signaling; clone: D5B10; catalog number: 13820; 1:100; incubation over night at 4 °C); CD31/PECAM-1 (R&D Systems; catalog number: AF3628; 1:100; 2 h at RT - room temperature); EMCN (Santa Cruz; clone V.5C7; catalog number: sc-65495; 1:100; 2 h at RT), COL10 (Abcam; catalog number: ab58632; 1:100; 2 h at RT); OSX (Abcam; catalog number: ab209484; 1:100; 2 h at RT); MMP9 Alexa Fluor 647 (Santa Cruz; clone: E-11; catalog number: sc-393859 AF647; 1:20; incubation over night at 4 °C); CD71-APC (BioLegend; clone: RI7217; catalog number: 113819; 1:100); Ter119-PE (BioLegend; catalog number: 116207; 1:100); CD31-PE (BioLegend; clone: MEC13.3; catalog number: 102507; 1:100); all secondary antibodies were purchased from Thermo Fisher Scientific and used at an 1:500 dilution for 2 h at RT if not stated otherwise: goat anti-rat A647 (A-21247), donkey anti-goat A568 (A-11057), goat anti-rat A488 (A-11006), goat anti-rabbit A647 (A-27040), goat anti-rabbit A488 (Abcam; ab150077; 1:1,000). DAPI (1:1,000; Sigma Aldrich) was added during the last washing step and sections were covered with Fluoromount_GT (Thermo Fisher Scientific), and a cover slip. Images were taken with a Keyence BZ9000

microscope (Keyence), a Zeiss LSM880 or an AxioScan (both Carl Zeiss Microscopy Deutschland GmbH) and image quantification was performed in a blinded manner using the Fiji/ImageJ software. The area of interest was manually created and managed with the built-in ROI-Manager. Arches, nuclei, lacuna and vascular loops were counted manually, and stained areas of interest (%) were determined with the thresholding tool. Definitions for quantification of vascular arches[34] and loops[22] are displayed in Supplementary Fig. 2.

## RNA analysis

For RNA analysis, left tibias from mice used for CECT analysis were treated with RNAlater (Qiagen) and stored at -80 °C until further use. Separation of the metaphysis and diaphysis was done by cutting underneath the growth plate. Bulk samples including bone and bone marrow were cryo-pulverized and resuspended in 1 ml ice-cold TriFast (VWR International) and carefully vortexed (30 s). A volume of 200 µl 1-bromo-3-chloropropane (Sigma Aldrich) was added and the mixture was incubated for 10 min at room temperature before centrifugation (10 min at 10.000 x g). The top aqueous phase was collected for RNA isolation using the RNeasy Mini Kit (Qiagen) following the manufacturer's instructions. Purity of the RNA was analyzed via Nanodrop; RNA integrity and quality were verified via Fragment Analyzer. cDNA synthesis was performed using the TaqMan Reverse Transcription Reagents (Applied Biosystems; 0.5 µg/µl RNA concentration) and DyNAmo Flash SYBR Green qPCR Kit (Thermo Fisher) was performed at a Stratagene Mx3000P (Agilent Technologies) with the following protocol: 7 min initial denaturation at 95 °C, 45–60 cycles of 10 s denaturation at 95 °C, 7 s annealing at 60 °C and 9 s elongation at 72 °C (duplicates per gene). CT values were normalized to *Hprt* (Housekeeper; ΔCt); as second control *18 s rRNA* was carried along (for internal validation only). The relative fold change expression was calculated using the following equation: $2^{-\Delta Ct}$. Primers were design using NCBI and Blast, tested and verified via Gel electrophoresis. Sequences are listed in Supplementary Table 1.

## Statistics and reproducibility

GraphPad Prism V.8 was used for statistical analysis. Data was tested for Gaussian distribution according to D'Agostino-Pearson omnibus normality test and homoscedasticity. When parametric test assumptions were met the Student's *t*-test (two-sided) was used to compare two groups. In case of failing normality testing, data were log-transformed, and residuals were evaluated prior to parametric testing on log-transformed data. A *p*-value < 0.05 was considered statistically significant. Sample sizes are indicated in the figure legends. Data are displayed with error bars showing mean ± SEM and individual samples in a bar graph. All analyses were performed on distinct samples. The following samples or data were excluded: Fig. 1—one sample in the SMAD1/5$^{WT}$ due to leakage in POM vessel staining and inconclusive distribution in the bone marrow, while bone parameters were still valid; Figs. 4 and 6—one sample from the *Emcn* RNA analysis in the SMAD1/5$^{iΔEC}$ group was deemed as outlier based on Tukey test. No further data was excluded from analysis.

## Reporting summary

Further information on research design is available in the Nature Portfolio Reporting Summary linked to this article.

## Data availability

All data supporting the findings of this study are available within the paper, its Supplementary Information and Supplementary Data 1. Raw image data are not openly available due to reasons of sensitivity and are available from the corresponding author upon reasonable request.

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

## Acknowledgements

The authors would like to thank all members of the Zwijsen and Boerckel labs for constructive discussions. Further, we thank Carla Geerome for assistance with the CECT scanning. The study was supported by the German Research Foundation – Research Fellowships: BE 6338/1-1 (A.B.), LA 4007/2-1 (A.L.) and the following grants: KU Leuven C12_16_023 and C14_19_095 (A.Z.); NIH R01AR073809 and NIH P30 AR069619 (J.D.B.).

## Author contributions

A.L., A.B., A.Z., and J.D.B. conceived and supervised the research. A.B. designed and performed animal experiments. A.L., A.W. and J.C. performed analysis. G.K. and T.B. supervised and conceived CECT scans and generated the CESA. A.L., A.B., A.Z. and J. D.B. wrote the paper. All authors revised the manuscript.

## Competing interests

The authors declare no competing interests.

## Additional information

[1]Departments of Orthopaedic Surgery and Bioengineering, University of Pennsylvania, Philadelphia, PA 19104, USA. [2]Department of Rheumatology and Clinical Immunology, Charité-Universitätsmedizin Berlin, corporate member of Freie Universität Berlin, Humboldt-Universität zu Berlin, and Berlin Institute of Health, Berlin 10117, Germany. [3]Department of Cardiovascular Sciences, Center for Molecular and Vascular Biology, KU Leuven, Leuven 3000, Belgium. [4]VIB-KU Leuven Center for Brain & Disease Research, KU Leuven, Leuven 3000, Belgium. [5]Department of Veterinary Medicine, Institute of Animal Welfare, Animal Behavior and Laboratory Animal Science, Freie Universität Berlin, Berlin 14163, Germany. [6]Institute of Mechanics, Materials and Civil Engineering, Biomechanics lab, UCLouvain, Louvain-la-Neuve 1348, Belgium. [7]Institute of Experimental and Clinical Research, Pole of Morphology, UCLouvain, Brussels 1200, Belgium. [8]KU Leuven, Department of Chemistry, Sustainable Chemistry for Metals and Molecules, Leuven 3000, Belgium. [9]Department of Materials Engineering, KU Leuven, Heverlee 3001, Belgium. [10]Division for Skeletal Tissue Engineering, Prometheus, KU Leuven, Leuven 3000, Belgium. [11]Present address: Centre for Translational Bone, Joint and Soft Tissue Research, Faculty of Medicine and University Hospital Carl Gustav Carus, Technische Universität Dresden (TUD), Fetscherstrasse 74, Dresden 01307, Germany.
✉e-mail: annemarie.lang@tu-dresden.de; boerckel@pennmedicine.upenn.edu

