## [Peer review file · Communications Biology]

Reviewers' comments:

Reviewer #1 (Remarks to the Author):

The authors of the manuscript tried to define the role of SMAD1/5 signaling in endothelial cells during endochondral ossification. To address this question, the authors used endothelial cell-specific Smad1/5 double knockout mice. Better understanding of angiogenic-osteogenic coupling is definitely very important and clinically relevant topic. However, in my opinion, there are several significant and conceptual issues of the study that have to be addressed.

It is known that SMAD1/5 signaling is crucial and fundamental for proper angiogenesis and vascularization in general, independent on the particular organ, as shown by authors in previously published papers using embryos and mouse retina (Moya et al., 2012, Benn et. al., 2020). The results of the current study are very similar to the ones that the authors have already presented in 2020, where vascularization of retina was used as a model to study the role of BMP-SMAD1/5 signaling in angiogenesis (Benn et. al., 2020). The authors also mention this fact in the discussion. Taking into account the general importance of this signaling for proper vessel formation, it is not surprising that the vessels in bone were also affected by the deletion of Smad1/5 in endothelial cells. Therefore, determining the role of SMAD1/5 signaling in particularly type H vessel morphogenesis at the very generalized level, that is used in the study, is redundant in terms of bringing useful/novel scientific knowledge. To make the main claims of the manuscript more justified, the type H specific properties have to be properly assessed that will provide better insight into the observed phenotype:

1. Definition of type H and L vessels is somehow misleading and it has to be improved to be clearer for the reader. Type H vessels are highly angiogenic blood vessel subtype that pave the way for surrounding osteoblasts during endochondral ossification (growth or regeneration). It is known that during peri-adolescence (5-6 weeks) type H vessels get replaced with quiescent type L vessels that do not exhibit bone growth promoting properties (Kusumbe et al., 2014; Dзамukova et al., 2022). In adult mice, vessels that are present both in bone marrow and in the quiescent ossification front/metaphysis are called type L vessels. They do not have columnar structure and their tip cells do not secrete MMP9 to digest the cartilage matrix. Therefore, it is not 100% correct to name the vessels "type H" in 7-8 week old mice (page 14), when there are mostly type L vessels left. In my view, it is more relevant for the current study to divide the vessels into bone-associated and marrow-associated.
2. Does the Smad1/5 deficiency in endothelial cells starting from 3 weeks lead to the differences in total length of long bones (femur, tibia)? Bone elongation is the primary task of type H vessels, so it is important to know, whether this parameter is affected.
3. Authors found that the number of anastomotic arches in type H vessels was reduced upon Smad1/5 deletion. It is known that non-resorbing vessel associated osteoclasts (VAOs) are responsible for anastomoses of type H vessels (Romeo et al., 2019). Therefore, the assessment, whether Smad1/5 deficiency in ECs leads to any differences in VAOs (at least, location and number) is essential.
4. Authors say that the columnar structure was impaired (page 7, line 139) upon EC-specific Smad1/5 deletion. However, by looking at the Figure 3a,b, one cannot say that the vessels in the metaphysis of SMAD1/5-deficient bones exhibit less columnar structure.
5. In the manuscript, authors assess the OSX/EMCN ratio. What is the biological meaning of such an assessment? The assessment of RUNX2/OSX ratio to evaluate the effect of endothelial SMAD1/5 depletion on osteoprogenitor dynamics would be more relevant.

6. The authors used bulk sequencing of the whole metaphysis or diaphysis to assess the expression of *Emcn*, *Sp7*, *Pecam*, *Dll4*, *Bmp6* and other genes. This approach is too unspecific and it does not provide information on the expression level in individual cells. Using multiplex RNAScope would be more appropriate to ascertain the expression level of particular genes in individual capillary. Authors claim that *Emcn* expression was upregulated in the metaphysis (Figure 4f) but the fold change of 1.2 on mRNA level is normally not considered as biologically significant (the standard threshold for differentially expressed genes is log fold change (\log_2) > 1). This upregulation could also be because of the higher density of vessels in *Smad1/5* deficient mice instead of higher EMCN in every individual endothelial cells (the same holds true for all other targets). Therefore, the visualization of the expression of these genes is necessary to make the conclusion on the role of SMAD1/5 in type H or type L EC-signaling.

7. On page 10, lines 182-184, the authors say that anastomotic arches of the type H vessels are responsible for the active degradation of cartilage matrix, which is not true. In the cited paper (Romeo et al., 2019), it is shown that the tip cells of type H endothelium (not arches) secrete MMP9 into the cartilage matrix to digest and invade it, which is not properly assessed in the study (only at very general mRNA level using bulk sequencing of the whole metaphysis). Conclusion on the role of SMAD1/5 signaling in cartilage resorption is not justified based on the presented results (page 10, lines 200-201). The observed phenotype with expanded hypertrophic zone of the growth plate upon EC-specific *Smad1/5* deficiency could be a result of abnormal vascularization, when the formation of tip and stalk cells is disrupted, as shown by the authors in the previous studies. Therefore, lack of proper cartilage resorption would be an indirect effect of corrupted angiogenesis. The staining for VEGFR3 and MMP9, that will visualize the cells with the tip properties, is essential for better understanding of the phenotype of endothelial cells in metaphysis.

8. What is the biological meaning of quantifying the EMCN/CD31 ratio and EMCN-positive and CD31-positive areas separately? It was shown before that both type H vessels and type L are CD31/EMCN double positive (Kusumbe et al., 2014; Ramasamy et al., 2014). How do authors explain that some ECs are not double positive? Can it be just an artefact because of using thin sections (the golden standard in type H vessel studies is to use thick (100-300 μm) sections for proper visualization of vessel network). Moreover, it is known that in the bone marrow monocytes (and other hematopoietic cells) can be also CD31-positive, which is not taken into account in the manuscript.

References

1. Benn, A.; Alonso, F.; Mangelschots, J.; Génot, E.; Lox, M.; Zwijsen, A. BMP-SMAD1/5 Signaling Regulates Retinal Vascular Development. *Biomolecules* 2020, 10, 488. <https://doi.org/10.3390/biom10030488>
2. Moya IM, Umans L, Maas E, Pereira PN, Beets K, Francis A, Sents W, Robertson EJ, Mummery CL, Huylebroeck D, Zwijsen A. Stalk cell phenotype depends on integration of Notch and *Smad1/5* signaling cascades. *Dev Cell*. 2012 Mar 13;22(3):501-14. doi: 10.1016/j.devcel.2012.01.007. Epub 2012 Feb 23. PMID: 22364862; PMCID: PMC4544746.
3. Kusumbe AP, Ramasamy SK, Adams RH. Coupling of angiogenesis and osteogenesis by a specific vessel subtype in bone. *Nature*. 2014 Mar 20;507(7492):323-328. doi: 10.1038/nature13145. Epub 2014 Mar 12. Erratum in: *Nature*. 2014 Sep 25;513(7519):574. PMID: 24646994; PMCID: PMC4943525.
4. Dzamukova, M., Brunner, T.M., Miotla-Zarebska, J. et al. Mechanical forces couple bone matrix mineralization with inhibition of angiogenesis to limit adolescent bone growth. *Nat Commun* 13, 3059 (2022). <https://doi.org/10.1038/s41467-022-30618-8>
5. Romeo SG, Alawi KM, Rodrigues J, Singh A, Kusumbe AP, Ramasamy SK. Endothelial proteolytic

activity and interaction with non-resorbing osteoclasts mediate bone elongation. *Nat Cell Biol.* 2019 Apr;21(4):430-441. doi: 10.1038/s41556-019-0304-7. Epub 2019 Apr 1. PMID: 30936475.

6. Ramasamy SK, Kusumbe AP, Wang L, Adams RH. Endothelial Notch activity promotes angiogenesis and osteogenesis in bone. *Nature.* 2014 Mar 20;507(7492):376-380. doi: 10.1038/nature13146. Epub 2014 Mar 12. PMID: 24647000; PMCID: PMC4943529.

Reviewer #2 (Remarks to the Author):

This study focuses on the effect of SMAD1/5 deletion during a specific post-natal window in endothelial cells (cells expressing cadherin 5). Vasculature, osterix-expressing cells, and bone and growth plate morphology were evaluated. Mutant animals showed hypervascularity and hyperpermeability in metaphyseal and diaphyseal regions as well as loss of osteoprogenitors and decreased cortical area and porosity. The authors conclude that SMAD1/5 signaling in endothelial cells regulates vascular morphogenesis, osteo-angio coupling, and bone formation

This study has broad, important implications for the field of bone development and homeostasis. The relationship between osteogenesis and angiogenesis is an area of intense investigation. This study points towards a mechanistic link involved in this relationship and will likely be a “jumping off point” for future work by multiple groups.

The strengths of this study are the separate analyses of the metaphyseal vs. diaphyseal regions, the technical quality of the immunohistological sections, and the clarity of the written and graphical presentations of the work.

The weaknesses are primarily in (1) the quantification of the vascular morphology; (2) the speculation about what is impacting bone growth; (3) the lack of examination of the fate of the osteoprogenitors.

1. There is no information provided on what defines an anastomotic arch ; (3) or a “large vascular loop” and how they were quantified, and the dense, amorphous appearance of the vasculature in figure 1B suggests non-specific staining. The conclusion that the mutant animals have hyperpermeable vessels deepens my concern about non-specificity. The emphasis of the text on morphology of the vasculature is not commensurate with the lack of quantification of important metrics of morphology, such as branch length, directionality, and connectivity. Related to this point the appearance of pronounced branching and network formation (lines 251-253) is not apparent in Figure 7, nor is the quantification of these characteristics described.

2. Statements such as “these findings...highlight the importance of vessel morphogenesis on long bone growth” (lines 123-4) are problematic because of the lack of deep quantification of morphogenesis and the lack of measurements of bone growth. Beyond the findings of altered microstructure in the metaphysis, diaphysis and growth plate, were there any differences in bone size (length or width)?

3. Although the lack of osterix+ cells is striking, there is no examination of what cell types are present in the regions where osterix+ cells are present in the WT mice. Is there any evidence of cell death? What is the morphology of the cells in this region? This additional analysis would add a nice level of mechanistic investigation to the work.

Specific comments

1. The values of cortical porosity shown in Figure 2i, ~40-60%, seem unreasonably high. For

comparison, <https://www.ncbi.nlm.nih.gov/pmc/articles/PMC3970724/> reports values of only ~10% for one-month-old C57BL/6 mice. Please provide information on how cortical porosity was measured, how this method was validated, and what guides were used to determine how reasonable the obtained values are.

2. Line 50: please provide one or more citations
3. Line 104: how were shaping vs. maintaining distinguished in this study?
4. The references to the figures are not all correct. Please check.
5. Figure 4i and 5b: please include a larger field of view
6. Figure 4 caption: what defines "active"?
7. Line 306: Reference 29 is not the primary source

Reviewer #3 (Remarks to the Author):

In this study by Lang et al., the authors describe the blood vessel and bone phenotypes of endothelial cell conditional Smad 1/5 depleted mice. They find that Smad1/5 regulates blood vessels in the metaphysis and diaphysis and changes in cortical bone. However, the study does not delve deep into any phenotypes to describe how Smad 1/5 regulates this phenotype. Moreover, some phenotypes need to be explained, and the study requires further analysis to consolidate and rationalise the observations.

1. The initial part of the study involves blood vessel analysis using (CECT) analysis. The later part of the study involves the analysis of microscopic images. However, they compare the phenotypes of both these analyses. For the readers to understand the connection and relation between both phenotypes, authors need to perform both analyses or at least a microscopic analysis of the initial phenotypes so that it becomes easier to understand the later part of the study.
2. The authors mention type H vessels in the manuscript, but they have not been exclusively quantified. Individual quantifications of Emcn+ area and CD31+ area and their area ratio would not indicate Emcnhi/ CD31hi subsets. It needs specific quantification of double-positive or Emcn & CD31 overlapping regions. The authors do not explain the importance of the CD31/Emcn ratio. What do they infer? What do the readers been advised about this ratio?
3. The authors do not explain the reason for the increase in cortical bone parameters. Type H vessels form a central part of the study as this subtype is specifically involved in angiogenesis & osteogenesis coupling. The quantification of CD31hi/Emcnhi (type H vessels is required to understand the data and interpret bone phenotypes. Authors have used Emcn+ area, CD31+ area and their ratio for blood vessel quantifications. There is a need to perform CD31hi/Emcnhi area quantifications. It is difficult to understand what the authors want to say regarding the increase in Emcn levels in protein and transcripts. They have compared these levels to changes in the Emcn+ area (pg7, line 151-152). Similarly, Pecan levels data and CD31+ area need explanation and what readers need to understand from the data. Is there an increase in typeH vessels? Is the increase in emcn levels without change in CD31 indicate angiogenesis? How do authors want readers to connect this data with cortical bone changes?
4. Vascular loop structures were mentioned to be present in the diaphysis, but supportive data were not provided to explain what these structures are. Are they similar to metaphyseal loops? Then it needs further explanation and high magnification 3D images to show the similarity. Or do the authors just want to say the vessels are tortuous? It needs an explanation. It is unclear and could mislead readers to conclude the formation of typeh vessels and loops in the diaphysis.
5. Cortical bone changes and their relation to blood vessels are interesting but need supportive

explanations of how it is manifested, the reason and the interaction

6. Pg7 line 151, 152 comparisons of Emcn+ area with Emcn transcript levels: Emcn+ area is not statistically increased, but transcripts are.

7. Pg10, there is inconsistency in the text description and data in figure 5..Please check figure 5

8. On line 189, cell morphology quantification data not provided

9. Figure 5d should show anastomosis data. No data provided. Also needs quantification.

10. The subtitle 'Endothelial SMAD1/5 signaling regulates endomucin expression and vascular maturation in diaphyseal sinusoidal (type L) capillaries' does not justify the text and data provided in this section. There is no data on vascular maturation and Emcn regulation. Did authors check Emcn levels in endothelial cells after activation of Smad1/5. Subtitle needs to be amended or supportive data has to be provided.

11. Pg 12, line 217, give reference for sprouting angiogenesis of type L and coupling with hematopoiesis.

12. Pg12, Vascular loops need to be explained. Should include Type h quantification.see comment 3.

13. Authors mention tip/stalk cells and sprouting angiogenesis but there is no data provided that supports the claim. Authors need to provide data to support the presence of sprouting angiogenesis in type L vessels.

14. Concluding sprouting angiogenesis based on total bone marrow Dll4 mRNA levels is biased. Dll4 is also expressed by hematopoietic cells in the marrow. An increase in Dll4 levels could be associated with the change in the hematopoietic compartment, which forms the majority of cell fraction in the marrow.

15. Vascular permeability- bone sinusoids (type L vessels) are fenestrated and do not have barriers. So erythrocyte immunostaining does not indicate vascular permeability. Did authors check permeability by any other methods? Is there any change in erythropoiesis?

16. Pg 14, line 252. High mag images and quantification data need to show 'branching and network formation' (figure 7b)

17. Supplementary fig1. Labels are missing - wt and cre+ images

Point-by-point Response to Reviewers

Dear Dr. Rauner & anonymous reviewers:

Thank you for your handling of our manuscript, for the opportunity to address the reviewers' feedback with this revision, and for the extension as we finalized experiments. We would like to especially thank the reviewers for their time and for their insightful and helpful feedback, which we believe has substantially elevated the rigor and the clarity of our manuscript. We have addressed the reviewers' questions with additional experiments, analysis and quantification, and edits to the manuscript, and we are excited to resubmit this revision. We have noted the changes we have made below. Please find included both a copy of the manuscript with all changes tracked as well as a clean copy for easier reading. Please find the reviewers' comments in *blue* followed by our responses (page numbers refer to the copy with changes tracked) below:

Response to Reviewer #1

The authors of the manuscript tried to define the role of SMAD1/5 signaling in endothelial cells during endochondral ossification. To address this question, the authors used endothelial cell-specific Smad1/5 double knockout mice. Better understanding of angiogenic-osteogenic coupling is definitely very important and clinically relevant topic. However, in my opinion, there are several significant and conceptual issues of the study that have to be addressed.

We would like to thank the reviewer for the helpful review, and for the suggestions that have greatly enhanced the clarity of our manuscript. We strongly believe that the revised manuscript is strengthened as a result and are grateful for the reviewer's contribution to our manuscript. We have addressed the reviewer's comments in a point-wise manner below, indicating how the manuscript has been changed in our response.

It is known that SMAD1/5 signaling is crucial and fundamental for proper angiogenesis and vascularization in general, independent on the particular organ, as shown by authors in previously published papers using embryos and mouse retina (Moya et al., 2012, Benn et. al., 2020). The results of the current study are very similar to the ones that the authors have already presented in 2020, where vascularization of retina was used as a model to study the role of BMP-SMAD1/5 signaling in angiogenesis (Benn et. al., 2020). The authors also mention this fact in the discussion. Taking into account the general importance of this signaling for proper vessel formation, it is not surprising that the vessels in bone were also affected by the deletion of Smad1/5 in endothelial cells. Therefore, determining the role of SMAD1/5 signaling in particularly type H vessel morphogenesis at the very generalized level, that is used in the study, is redundant in terms of bringing useful/novel scientific knowledge. To make the main claims of the manuscript more justified, the type H specific properties have to be properly assessed that will provide better insight into the observed phenotype:

Indeed, we (An Zwijsen and team) have previously shown the importance of SMAD1/5 signaling in vascular development with a particular focus on the embryo and early post-natal development (Moya et al. 2012 – hindbrain E9.5; Benn et al. 2020 – retina P4/P6). We demonstrated previously that during mouse embryonic development, Notch and SMAD1/5 signaling synergize to balance selection of tip and stalk cells in vascular sprouting. Kusumbe et al. 2014 impressively described the morphology of the bone/bone marrow vasculature of long bones and underlined the specificity of this vascular network when compared with other organs. Ramsamy et al. 2014 further reported that Notch signaling, in particular, was important for osteo-angiogenic coupling and showed the consequence on bone formation. Therefore, in our study we investigated the role of SMAD1/5 activity in the endothelium during juvenile age with a particular interest in the bone vasculature and the potential role of endothelial SMAD1/5 activity in bone formation/osteo-angiogenic coupling. To our knowledge this is the first study particularly focusing on this question.

1. Definition of type H and L vessels is somehow misleading, and it has to be improved to be clearer for the reader. Type H vessels are highly angiogenic blood vessel subtype that pave the way for surrounding osteoblasts during endochondral ossification (growth or regeneration). It is known that during peri-adolescence (5-6 weeks) type H vessels get replaced with quiescent type L vessels that do not exhibit bone growth promoting properties (Kusumbe et al., 2014; Dzamukova et al., 2022). In adult mice, vessels that are present both in bone marrow and in the quiescent ossification front/metaphysis are called type L vessels. They do not have columnar structure and their tip cells do not secrete MMP9 to digest the cartilage matrix. Therefore, it is not 100% correct to name the vessels “type H” in 7-8 week old mice (page 14), when there are mostly type L vessels left. In my view, it is more relevant for the current study to divide the vessels into bone-associated and marrow-associated.

We thank the reviewer for raising these important points. We have changed our wording to reflect the identity of our vessels of interest as “metaphyseal” and “diaphyseal” vessels, rather than “Type H” and “Type L,” respectively. As we did not flow-sort for endothelial cells based on their expression levels of CD31 and Endomucin, which is the established method to define the “type H” and “type L” vessels, but rather direct our analysis based on the anatomical location and morphology, we have removed these labels.

2. Does the *Smad1/5* deficiency in endothelial cells starting from 3 weeks lead to the differences in total length of long bones (femur, tibia)? Bone elongation is the primary task of type H vessels, so it is important to know, whether this parameter is affected.

We thank the reviewer for this important point; however, we have not examined long bone growth *per se*, as we do not expect differences within 1 or 2 weeks of endothelial *Smad1/5* deletion. Further, our high-resolution microCT data reveals no significant differences in the metaphyseal bone formation at this time point. Although we see significant differences in metaphyseal bone formation, at 2 weeks post-tamoxifen (P35; see new added panel in **Fig. 4j**, page 10), we do not expect tremendous differences in growth. The timepoint/age was chosen as it represents a period of rapid bone developmental adaptations necessary to maintain and support long bone growth. We have removed the words “during bone growth” from our discussion and from the title of the paper to clarify that this paper is not focused on the effects on limb elongation, but rather on local angiogenic-osteogenic crosstalk at shorter time scales.

Figure 4j. Quantification of BV/TV – quantitative μCT analysis (P35; $n^{WT} = 16$; $n^{\Delta EC} = 4$).

3. Authors found that the number of anastomotic arches in type H vessels was reduced upon *Smad1/5* deletion. It is known that non-resorbing vessel associated osteoclasts (VAOs) are responsible for anastomoses of type H vessels (Romeo et al., 2019). Therefore, the assessment, whether *Smad1/5* deficiency in ECs leads to any differences in VAOs (at least, location and number) is essential.

We agree with the reviewer that quantification of VAOs might help explain the reduced arch numbers in the metaphyseal vessels at the chondro-osseous junction. However, since we have chosen a different methodological approach in terms of section thickness, VAO staining could be unprecise these cells are present in relatively low numbers. To address the potential mechanisms of altered growth plate resorption, independent of the cell types responsible, we have now added staining for MMP9. Romeo et al. showed impressively that MMP9 produced by the endothelial cells themselves results in tremendous growth plate enlargement (cf. Fig. 4 and 5; Romeo et al. 2019). We found markedly reduced MMP9 expression associated to the EMCN⁺ vasculature by P35. The following changes have been made:

- Results (page 12):

It has been shown that MMP9 activity in metaphyseal vessels directs growth plate size during bone development ²⁶. Therefore, we performed additional MMP9 staining in the metaphysis and found markedly reduced metaphyseal MMP9 abundance by P35, 14 days post-tamoxifen (Supplementary Figure 5).

- Supplementary Figure 5:

Supplementary Figure 5. MMP9 and EMCN staining in the metaphysis at 14 days post-tamoxifen injection (P35). Representative images displaying two independent samples (n= 2) per group. Scale bars indicate 100 μ m.

- Discussion (page 20):

Therefore, the dysmorphogenesis of the metaphyseal vessels at the chondro-osseous junction upon EC-specific SMAD1/5 depletion could be the reason for the enlargement of the hypertrophic chondrocyte zone, as a result of reduced expression of MMP9 in the metaphyseal vasculature.

4. Authors say that the columnar structure was impaired (page 7, line 139) upon EC-specific *Smad1/5* deletion. However, by looking at the Figure 3a,b, one cannot say that the vessels in the metaphysis of *SMAD1/5*-deficient bones exhibit less columnar structure.

We appreciate this corrective and have removed such statements.

5. In the manuscript, authors assess the *OSX/EMCN* ratio. What is the biological meaning of such an assessment? The assessment of *RUNX2/OSX* ratio to evaluate the effect of endothelial *SMAD1/5* depletion on osteoprogenitor dynamics would be more relevant.

We appreciate that the usage of these ratios was confusing, so we have removed them from the manuscript.

6. The authors used bulk sequencing of the whole metaphysis or diaphysis to assess the expression of *Emcn*, *Sp7*, *Pecam*, *Dll4*, *Bmp6* and other genes. This approach is too unspecific and it does not provide information on the expression level in individual cells. Using multiplex *RNA*Scope would be more appropriate to ascertain the expression level of particular genes in individual capillary. Authors claim that *Emcn*

expression was upregulated in the metaphysis (Figure 4f) but the fold change of 1.2 on mRNA level is normally not considered as biologically significant (the standard threshold for differentially expressed genes is log fold change (log2) > 1). This upregulation could also be because of the higher density of vessels in *Smad1/5* deficient mice instead of higher *EMCN* in every individual endothelial cells (the same holds true for all other targets). Therefore, the visualization of the expression of these genes is necessary to make the conclusion on the role of *SMAD1/5* in type H or type L EC-signaling.

Using RNA expression data from bulk tissue can be indeed unprecise – however, we have used the gene expression data only in selected occasions to underline certain aspects/findings already assayed spatially by immunofluorescent protein labeling to provide an additional level of evidence rather than trying to fully establish mechanistical relations based on these data. Since we do not provide protein data for *Dll4* and *Bmp6*, we have removed this data from the results, as well as *Pecam*. However, we disagree that the *Emcn* mRNA expression is irrelevant, particularly when taken together with the *Id1* and *Sp7* gene expression data. For analysis and visualization, CT values were normalized to the housekeeper gene (*Hprt*) and z-score normalization was performed after calculating the relative fold change expression ($2^{-\Delta Ct}$). We understand that this additional normalization might be misleading when the pure fold change is of interest. **Thus, we have now adapted the data to display the relative fold change expression ($2^{-\Delta Ct}$).** The following changes have been made:

- Figure 4d-f (page 10):

Figure 4. Relative mRNA expression of (d) *Id1*, (e) *Emcn* and (f) *Sp7* normalized to *Hprt* (housekeeping gene) in the epi-/metaphysis (P28; n= 10).

- Results (page 8):

As expected, *Id1* expression was significantly lower in the meta-/epiphysis of *SMAD1/5*^{iΔEC} mice ($p = 0.03$; **Fig. 4d**). Although *EMCN*⁺ area was not significantly elevated (**Fig. 3d**), EC-specific depletion of *SMAD1/5* increased *Emcn* mRNA abundance (fold change between means 1.5; $p = 0.02$; **Fig. 4e**). *Sp7* (*OSX*) mRNA was also significantly reduced (44% lower, $p = 0.04$) by endothelial *SMAD1/5* deactivation at P28 (**Fig. 4f**).

- Figure 6f (page 15):

Figure 6f. Relative mRNA expression analysis of *Emcn* normalized to *Hprt* (housekeeping gene) in the diaphysis (P28; n= 10).

- Material and Methods (page 17):

CT values were normalized to *Hprt* (Housekeeper; ΔCt); as second control *18s rRNA* was carried along (for internal validation only). The relative fold change expression was calculated using the following equation: $2^{-\Delta Ct}$.

7. On page 10, lines 182-184, the authors say that anastomotic arches of the type H vessels are responsible for the active degradation of cartilage matrix, which is not true. In the cited paper (Romeo et al., 2019), it is shown that the tip cells of type H endothelium (not arches) secrete MMP9 into the cartilage matrix to digest and invade it, which is not properly assessed in the study (only at very general mRNA level using bulk sequencing of the whole metaphysis). Conclusion on the role of SMAD1/5 signaling in cartilage resorption is not justified based on the presented results (page 10, lines 200-201). The observed phenotype with expanded hypertrophic zone of the growth plate upon EC-specific Smad1/5 deficiency could be a result of abnormal vascularization, when the formation of tip and stalk cells is disrupted, as shown by the authors in the previous studies. Therefore, lack of proper cartilage resorption would be an indirect effect of corrupted angiogenesis. The staining for VEGFR3 and MMP9, that will visualize the cells with the tip properties, is essential for better understanding of the phenotype of endothelial cells in metaphysis.

We thank the reviewer for this comment and agree that the wording regarding the anastomotic arches is misleading. We have revised the respective sentences as outlined in the following:

- Results (page 12):
The metaphyseal capillaries function not only to support network connectivity and osteoprogenitor mobilization, but together with surrounding cells ²⁶, actively degrade the hypertrophic cartilage to enable endochondral ossification ²⁶.
- Discussion (page 20):
Therefore, the dysmorphogenesis of the metaphyseal vessels at the chondro-osseous junction upon EC-specific SMAD1/5 depletion could be the reason for the enlargement of the hypertrophic chondrocyte zone, as a result of reduced expression of MMP9 in the metaphyseal vasculature.

In addition, as explained above (question 3), we have now added staining for MMP9 (Supplementary Fig. 5) and found markedly reduced MMP9 expression associated to the EMCN+ vasculature by P35.

8. What is the biological meaning of quantifying the EMCN/CD31 ratio and EMCN-positive and CD31-positive areas separately? It was shown before that both type H vessels and type L are CD31/EMCN double positive (Kusumbe et al., 2014; Ramasamy et al., 2014). How do authors explain that some ECs are not double positive? Can it be just an artefact because of using thin sections (the golden standard in type H vessel studies is to use thick (100-300 μm) sections for proper visualization of vessel network). Moreover, it is known that in the bone marrow monocytes (and other hematopoietic cells) can be also CD31-positive, which is not taken into account in the manuscript.

As we did not flow-sort for endothelial cells based on their expression levels of CD31 and Endomucin (established method to define the “type H” and “type L” vessels) and chose a different methodological approach in terms of section thickness, our analysis is particularly focused on the anatomical location and morphology and the respective wording in the manuscript has been adapted. We note that Endomucin and CD31 do not mark exactly the same cell populations – for example, CD31 is also expressed in a subset of myeloid-lineage cells. Thus, while we are not specifically querying changes to the *Emcn*^{hi} CD31^{hi} vessels, as described in the literature, we do use these two standard but non-overlapping markers of the endothelium to accurately visualize and quantify vascular morphology. Finally, we appreciate the reviewers’ input that the presentation of expression ratios is confusing – we have therefore removed these parameters.

Response to Reviewer #2

This study focuses on the effect of SMAD1/5 deletion during a specific post-natal window in endothelial cells (cells expressing cadherin 5). Vasculature, osterix-expressing cells, and bone and growth plate morphology were evaluated. Mutant animals showed hypervascularity and hyperpermeability in metaphyseal and diaphyseal regions as well as loss of osteoprogenitors and decreased cortical area and porosity. The authors conclude that SMAD1/5 signaling in endothelial cells regulates vascular morphogenesis, osteo-angio coupling, and bone formation

This study has broad, important implications for the field of bone development and homeostasis. The relationship between osteogenesis and angiogenesis is an area of intense investigation. This study points towards a mechanistic link involved in this relationship and will likely be a “jumping off point” for future work by multiple groups.

The strengths of this study are the separate analyses of the metaphyseal vs. diaphyseal regions, the technical quality of the immunohistological sections, and the clarity of the written and graphical presentations of the work.

The weaknesses are primarily in (1) the quantification of the vascular morphology; (2) the speculation about what is impacting bone growth; (3) the lack of examination of the fate of the osteoprogenitors.

We would like to thank the reviewer for the helpful review, and for the suggestions which have strengthened our manuscript. We have addressed the reviewer's comments in a point-wise manner below with particular focus on providing more rationales on our analysis of the vascular morphology, clarifying the wording “bone growth” and addressing potential future avenues to examine the fate of the osteoprogenitors.

1. There is no information provided on what defines an anastomotic arch ; (3) or a “large vascular loop” and how they were quantified, and the dense, amorphous appearance of the vasculature in figure 1B suggests non-specific staining. The conclusion that the mutant animals have hyperpermeable vessels deepens my concern about non-specificity. The emphasis of the text on morphology of the vasculature is not commensurate with the lack of quantification of important metrics of morphology, such as branch length, directionality, and connectivity. Related to this point the appearance of pronounced branching and network formation (lines 251-253) is not apparent in Figure 7, nor is the quantification of these characteristics described.

We thank the reviewer for noting the need for increased clarity. In the following we would like to address:

1. the missing definitions or explanations regarding the morphometric analysis and
2. the exclusion of non-specific stainings.

Comment on 1.: We have now expanded our materials and methods to describe our morphometric process for the anastomotic arches and the vascular loops, and we have included a detailed figure in the Supplementary Information for clarity. We note that the bone marrow vasculature organization and morphology is unique and is not amenable to common vascular tree parameters (branch length, directionality, connectivity) as typically used in, for example, retinal angiogenesis assays. We have therefore used the standard parameters established in the field by the Adams, Kusumbe, and Ramasamy groups and others for analysis of bone marrow and bone-associated vasculature. The following changes have been made:

- Material and Methods (page 25):
Arches, nuclei, lacuna and vascular loops were counted manually, and stained areas of interest (%) were determined with the thresholding tool. Definitions for quantification of vascular arches³⁰ and loops²² are displayed in **Supplementary Fig. 2.**

- Results (page 8):
To determine the role of postnatal SMAD1/5 signaling in metaphyseal vessel morphogenesis and angiogenic-osteogenic coupling, we first examined the number of vessel arches adjacent to the growth plate and the area of EMCN and CD31-expressing vessels in the metaphysis (**Supplementary Fig. 2**).
- Supplementary Figure 2

Supplementary Figure 2. Quantification of vascular arches and loops. (a) Vascular columns are linked by tubular arches next to the growth plate chondrocytes. (b) Vascular loops are defined by a closed enclosure lined with Emcn+ cells around a lumen. * Asterisk in schematics indicate exemplary representation of either arches (a) or loops (b). Arrows indicate either exemplary arches (a) or loops (b) in magnifications.

- Results (page 16) – removed “of pronounced branching and network formation”.

Comment on 2.: With POM-based CECT imaging of the marrow vasculature, the hematopoietic tissue is stained (based on electrostatic interactions) with the Tungsten-containing POM compound (established and described by Kerckhofs et al. 2018). It is therefore increased in X-ray attenuation but is still less attenuating than the bone. Since the vessels are empty (or only few red blood cells are present), there is no staining inside the blood vessels, and this is how the vessel lumen is visualized. Permeabilization or leaking does in the case of POMs as contrast agent does not play a role, as we do not perfuse the contrast agent through the vasculature. We have now included a more precise explanation:

- Material and Methods (page 23):
The Tungsten-containing POM compound therefore stains the hematopoietic tissue (based on electrostatic interactions) while blood vessels are empty with no staining which allows the subsequent visualization.

Kerckhofs, G. et al. Simultaneous three-dimensional visualization of mineralized and soft skeletal tissues by a novel microCT contrast agent with polyoxometalate structure. *Biomaterials* 159, 1-12, doi:10.1016/j.biomaterials.2017.12.016 (2018).

2. Statements such as “these findings...highlight the importance of vessel morphogenesis on long bone growth” (lines 123-4) are problematic because of the lack of deep quantification of morphogenesis and the lack of measurements of bone growth. Beyond the findings of altered microstructure in the metaphysis, diaphysis and growth plate, were there any differences in bone size (length or width)?

We thank the reviewer for identifying this lack of clarity in our manuscript. Our original intention, in using the phrase “long bone growth” was to indicate that we were analyzing the vasculature and bone morphometry at postnatal stages, during which bone growth was active. However, our purpose in conducting this study was not to assess the long-term impacts on bone growth, *per se*, but rather to assess the effects of short-term SMAD1/5 deletion on vascular morphology and local osteogenic coupling. To clarify this, we have changed the title from “Endothelial SMAD1/5 signaling couples angiogenesis to osteogenesis during bone

growth” to “Endothelial SMAD1/5 signaling couples angiogenesis to osteogenesis in juvenile bone”. We have also made the point explicit in the paper, removing references to “bone growth.”

3. Although the lack of osterix+ cells is striking, there is no examination of what cell types are present in the regions where osterix+ cells are present in the WT mice. Is there any evidence of cell death? What is the morphology of the cells in this region? This additional analysis would add a nice level of mechanistic investigation to the work.

We agree with the reviewer that this would be an excellent next direction for this project, and we are planning grant proposal submissions that would enable continued funding to conduct this mechanistic work. Unfortunately, this analysis is beyond the scope of the present study.

Specific comments

1. The values of cortical porosity shown in Figure 2i, ~40-60%, seem unreasonably high. For comparison, <https://www.ncbi.nlm.nih.gov/pmc/articles/PMC3970724/> reports values of only ~10% for one-month-old C57BL/6 mice. Please provide information on how cortical porosity was measured, how this method was validated, and what guides were used to determine how reasonable the obtained values are.

The reviewer is correct. The outcomes were selected and calculated based on Bouxein et al. which has been now explicitly stated in Materials and Methods (page 24):

- Relevant trabecular and cortical outcomes were selected and calculated following Bouxein et al.⁴⁹. Bouxein, M. L. et al. Guidelines for assessment of bone microstructure in rodents using micro-computed tomography. *Journal of Bone and Mineral Research* 25, 1468-1486

However, our current analysis pipeline does not allow to determine the porosity precisely, thus, we have removed this dataset from the manuscript.

2. Line 50: please provide one or more citations

We have added the following references:

Gerber, H.-P. et al. VEGF couples hypertrophic cartilage remodeling, ossification and angiogenesis during endochondral bone formation. *Nature Medicine* 5, 623-628, doi:10.1038/9467 (1999).

Maes, C. et al. Osteoblast Precursors, but Not Mature Osteoblasts, Move into Developing and Fractured Bones along with Invading Blood Vessels. *Developmental Cell* 19, 329-344, doi:10.1016/j.devcel.2010.07.010 (2010).

Collins, J. M. et al. YAP and TAZ couple osteoblast precursor mobilization to angiogenesis and mechanoregulated bone development. *bioRxiv*, 2023.2001.2020.524918, doi:10.1101/2023.01.20.524918 (2023).

3. Line 104: how were shaping vs. maintaining distinguished in this study?

We define shaping as a dynamic process as metaphyseal vessels have to adapt growth activities in the juvenile bone. In contrast, diaphyseal vessels are supposed to be rather quiescent and emerge from metaphyseal vessels (= maintenance). We have revised the following sentence in the *Discussion (page 20)*:

- Growth plate remodeling and endochondral ossification are mediated by metaphyseal vessels, but the long bone diaphysis is populated by sinusoidal vessels. These vessels have been proposed to be maintained in a homeostatic and quiescent state with relatively slower physiological remodeling⁷. Kusumbe et al. suggest that diaphyseal vessels emerge through maturation of metaphyseal capillaries

^{7,27}

4. The references to the figures are not all correct. Please check.

We like to thank the reviewer for careful reading – and we have now corrected the references to the figures.

5. Figure 4i and 5b: please include a larger field of view

We have changed the Figures accordingly:

- Figure 4h (page 10):

Figure 4h. Representative images of EMCN and OSX staining in the metaphysis (P35; n^{WT}= 4; n^{ΔEC}= 4).

- Figure 5b (page 13)

Figure 5b. Representative images of H&E staining at P28.

6. Figure 4 caption: what defines “active”?

We have removed the term “active”.

7. Line 306: Reference 29 is not the primary source

We have changed the whole paragraph – Discussion (pages 19/20):

Cortical bone was also impaired after short-term depletion of endothelial SMAD1/5 activity (P28). Although our data does not allow to draw conclusions on the mechanism which induce the changes in the cortical bone, we speculate that it might be a result of the changes in the metaphyseal bone or due to an impairment in transcortical vessels. In detail, it is known that the process of corticalization is impacted by metaphyseal bone formation and subsequent remodeling^{31,32}. In addition, cortical bone has been shown to have transcortical blood vessels which may impact cortical bone morphogenesis, though this has not been studied so far³³. Further studies are needed to define the mechanism for angiogenic-osteogenic coupling in the cortical bone during development and remodeling.

References:

- Colnot, C., Thompson, Z., Miclau, T., Werb, Z. & Helms, J. A. Altered fracture repair in the absence of MMP9. *Development* **130**, 4123-4133, doi:10.1242/dev.00559 (2003).
- Wang, Q., Ghasem-Zadeh, A., Wang, X. F., Iuliano-Burns, S. & Seeman, E. Trabecular bone of growth plate origin influences both trabecular and cortical morphology in adulthood. *J Bone Miner Res* **26**, 1577-1583, doi:10.1002/jbmr.360 (2011).
- Grüneboom, A. *et al.* A network of trans-cortical capillaries as mainstay for blood circulation in long bones. *Nat Metab* **1**, 236-250, doi:10.1038/s42255-018-0016-5 (2019).

Response to Reviewer #3

In this study by Lang et al., the authors describe the blood vessel and bone phenotypes of endothelial cell conditional Smad 1/5 depleted mice. They find that Smad1/5 regulates blood vessels in the metaphysis and diaphysis and changes in cortical bone. However, the study does not delve deep into any phenotypes to describe how Smad 1/5 regulates this phenotype. Moreover, some phenotypes need to be explained, and the study requires further analysis to consolidate and rationalise the observations.

1. The initial part of the study involves blood vessel analysis using (CECT) analysis. The later part of the study involves the analysis of microscopic images. However, they compare the phenotypes of both these analyses. For the readers to understand the connection and relation between both phenotypes, authors need to perform both analyses or at least a microscopic analysis of the initial phenotypes so that it becomes easier to understand the later part of the study.

We thank the reviewer for these helpful suggestions that clarify the paper and enable understanding by a broader audience. We have now included microscopic analyses for both short-term (7 days; P28) and long term (14 days; P35) SMAD1/5 depletion from the endothelium in the metaphysis (**Fig. 3**) and diaphysis (**Fig. 6**) of juvenile mice and provide additional microCT analysis of the metaphysis at P35 (**Fig. 4j**). The following changes have been made:

- Results (page 8):

For the experimental design, we chose both short-term (7 days) and long-term (14 days) SMAD1/5 depletion from the endothelium of juvenile mice. Thus, we investigated cellular changes at P28 (7d after first tamoxifen injection; **Fig. 3a**) and P35 (14 days after first tamoxifen injection; **Fig. 3f**). EC-specific SMAD1/5 depletion resulted in aberrant vascular architecture (**Fig. 3b**) and reduced the number of anastomotic arches (mean difference = 3.8 ± 1.2 arches/mm; $p = 0.02$) at the chondro-osseous junction (**Fig. 3c**) but did not significantly alter EMCN⁺ and CD31⁺ areas in the metaphysis (**Fig. 3d, e**). By two-weeks post tamoxifen, EC-specific SMAD1/5 depletion resulted in an aberrant columnar structure of the metaphyseal vessels and significant reduced anastomotic arch numbers ($p < 0.001$; **Fig. 3g, h**). EMCN⁺ and CD31⁺ areas in the metaphysis were not significantly reduced (**Fig. 3i, j**).

- Figure 3 (page 9): addition of quantifications f-g

Figure 3. Endothelial SMAD1/5 activity regulates to metaphyseal vessel morphology in juvenile mice. (a) Tamoxifen treatment and short-term sampling scheme. Mice were injected postnatal day 19-21 (P19-21) and samples were collected at P28. (b) Representative images of EMCN and CD31 staining in the tibial metaphyseal and diaphyseal area showing EMCN, CD31 and DAPI staining in the metaphyseal area (P28; * $n^{WT}=4$; # $n^{iΔEC}=6$). Quantification of (c) arch number, (d) EMCN⁺ and (e) CD31⁺ areas (P28; $n^{WT}=4$; $n^{iΔEC}=6$). (f) Tamoxifen treatment and long-term sampling scheme. Mice were injected postnatal day 19-21 (P19-21) and samples were collected at P35. (g) Representative images of EMCN and CD31 staining in the tibial metaphyseal area and magnifications showing EMCN, CD31 and DAPI staining (P35; * $n^{WT}=4$; # $n^{iΔEC}=4$). Quantification of (h) arch number, (i) EMCN⁺ and (j) CD31⁺ areas (P35; $n^{WT}=4$; $n^{iΔEC}=4$). gp – growth plate. Bar graphs show mean \pm SEM and individual data points. Two-sample t-test was used to determine the statistical significance; p-values are indicated with * $p < 0.05$. All scale bars indicate (b) 500 μ m, (g) 250 μ m and 125 μ m (magnifications).

- Figure 4i (page 10):

Figure 4j. Quantification of BV/TV – quantitative μ CT analysis (P35; $n^{WT}= 16$; $n^{i\Delta EC}= 4$).

- Results (page 8):

By P35, 14 days post-tamoxifen (Fig. 6g), EC-specific SMAD1/5 depletion resulted in more pronounced dysmorphogenesis of the diaphyseal vessels, which did not allow for vascular loop quantification due to loss of network integrity (Fig. 6h). EMCN⁺ area (Fig. 6i) and CD31⁺ area (Fig. 6j) were increased although not significantly.

- Figure 6 (page 9): addition of quantifications i, j

Figure 6. Endothelial SMAD1/5 promotes maturation and maintenance of diaphyseal sinusoidal capillaries. (a) Tamoxifen treatment and short-term sampling scheme. Mice were injected postnatal day 19-21 (P19-21) and samples were collected at P28. (b) Representative images of EMCN and CD31 staining in the diaphysis (P28; $n^{WT}= 4$; $n^{i\Delta EC}= 6$).

Quantification of (c) number of vascular loops per mm², (d) relative EMCN⁺ and (e) CD31⁺ area (P28; n^{WT}= 4; n^{iΔEC}= 6). (f) Relative mRNA expression analysis of *Emcn* normalized to *Hprt* (housekeeping gene) in the diaphysis (P28; n= 10). (g) Tamoxifen treatment and long-term sampling scheme. Mice were injected postnatal day 19-21 (P19-21) and samples were collected at P35. (h) Representative images of EMCN and CD31 staining in the diaphysis (P35; n = 4). Quantification of (i) relative EMCN⁺ and (j) CD31⁺ area (P35; n = 4). Bar graphs show mean ± SEM and individual data points. Two-sample t-test was used to determine the statistical significance; p-values are indicated with *p < 0.05; ***p < 0.001. All scale bars indicate 250 μm (b) or 125 μm (magnifications b, h).

*2. The authors mention type H vessels in the manuscript, but they have not been exclusively quantified. Individual quantifications of *Emcn*⁺ area and CD31⁺ area and their area ratio would not indicate *Emcn*^{hi}/*CD31*^{hi} subsets. It needs specific quantification of double-positive or *Emcn* & CD31 overlapping regions. The authors do not explain the importance of the CD31/*Emcn* ratio. What do they infer? What do the readers been advised about this ratio?*

We thank the reviewer for raising these important points. We have changed our wording to reflect the identity of our vessels of interest as “metaphyseal” and “diaphyseal” vessels, rather than “Type H” and “Type L,” respectively. As we did not flow-sort for endothelial cells based on their expression levels of CD31 and Endomucin, which is the established method to define the “type H” and “type L” vessels, but rather direct our analysis based on the anatomical location and morphology, we have removed these labels. We note that Endomucin and CD31 do not mark exactly the same cell populations – for example, CD31 is also expressed in a subset of myeloid-lineage cells. Thus, while we are not specifically querying changes to the *Emcn*^{hi} CD31^{hi} vessels, as described in the literature, we do use these two standard but non-overlapping markers of the endothelium to accurately visualize and quantify vascular morphology. Finally, we appreciate the reviewers’ input that the expression of expression ratios is confusing – we have therefore removed these parameters.

*3. The authors do not explain the reason for the increase in cortical bone parameters. Type H vessels form a central part of the study as this subtype is specifically involved in angiogenesis & osteogenesis coupling. The quantification of CD31^{hi}/*Emcn*^{hi} (type H vessels is required to understand the data and interpret bone phenotypes. Authors have used *Emcn*⁺ area, CD31⁺ area and their ratio for blood vessel quantifications. There is a need to perform CD31^{hi}/*Emcn*^{hi} area quantifications. It is difficult to understand what the authors want to say regarding the increase in *Emcn* levels in protein and transcripts. They have compared these levels to changes in the *Emcn*⁺ area (pg7, line 151-152). Similarly, *Pecam* levels data and CD31⁺ area need explanation and what readers need to understand from the data. Is there an increase in typeH vessels? Is the increase in *emcn* levels without change in CD31 indicate angiogenesis? How do authors want readers to connect this data with cortical bone changes?*

We thank the reviewer for the helpful identification of a lack of clarity in our paper regarding our measurement of vascular morphology in the various bone marrow compartments and its impact on trabecular and cortical bone. As explained in the response to comment 2, we have adapted the wording to “metaphyseal” and “diaphyseal” vessels, since our methodological approach does not allow a precise differentiation between “type H” and “type L” vessels. We are also aware that Endomucin and CD31 do not mark exactly the same cell populations – for example, CD31 is also expressed in endothelial progenitors and other myeloid-lineage cells, while Endomucin is a more reliable marker to analyze and quantify vascular morphology. We have now inserted an explanatory text in the Results section (page 8):

Postnatally, EMCN is a specific vascular marker for sinusoidal and venous vessels, while CD31 is additionally expressed by arteries, endothelial progenitors and other myeloid-lineage cells ⁷.

Regarding the connection/discussion of the cortical bone changes, we would like to refer to our more detailed response to comment 5.

4. Vascular loop structures were mentioned to be present in the diaphysis, but supportive data were not provided to explain what these structures are. Are they similar to metaphyseal loops? Then it needs further explanation and high magnification 3D images to show the similarity. Or do the authors just want to say the vessels are tortuous? It needs an explanation. It is unclear and could mislead readers to conclude the formation of type h vessels and loops in the diaphysis.

We have now expanded our materials and methods section to describe our morphometric process to analyze anastomotic arches and vascular loops, and we have included a detailed figure in the Supplementary Information for clarity. The following changes have been made:

- Material and Methods (page 25):
Arches, nuclei, lacuna and vascular loops were counted manually, and stained areas of interest (%) were determined with the thresholding tool. Definitions for quantification of vascular arches³⁰ and loops²² are displayed in **Supplementary Fig. 2**.
- Results (page 8):
To determine the role of postnatal SMAD1/5 signaling in metaphyseal vessel morphogenesis and angiogenic-osteogenic coupling, we first examined the number of vessel arches adjacent to the growth plate and the area of EMCN and CD31-expressing vessels in the metaphysis (**Supplementary Fig. 2**).
- Supplementary Figure 2

Supplementary Figure 2. Quantification of vascular arches and loops. (a) Vascular columns are linked by tubular arches next to the growth plate chondrocytes. (b) Vascular loops are defined by a closed enclosure lined with Emcn+ cells around a lumen. * Asterisk in schematics indicate exemplary representation of either arches (a) or loops (b). Arrows indicate either exemplary arches (a) or loops (b) in magnifications.

5. Cortical bone changes and their relation to blood vessels are interesting but need supportive explanations of how it is manifested, the reason and the interaction

We agree with the reviewer and have adapted the Discussion (pages 19/20) as follows:

Cortical bone was also impaired after short-term depletion of endothelial SMAD1/5 activity (P28). Although our data does not allow to draw conclusions on the mechanism which induce the changes in the cortical bone, we speculate that it might be a result of the changes in the metaphyseal bone or due to an impairment in transcortical vessels. In detail, it is known that the process of corticalization is impacted by metaphyseal bone formation and subsequent remodeling^{31,32}. In addition, cortical bone has been shown to have transcortical blood vessels which may impact cortical bone morphogenesis, though this has not been studied so far³³. Further studies are needed to define the mechanism for angiogenic-osteogenic coupling in the cortical bone during development and remodeling.

References:

- Colnot, C., Thompson, Z., Miclau, T., Werb, Z. & Helms, J. A. Altered fracture repair in the absence of MMP9. *Development* **130**, 4123-4133, doi:10.1242/dev.00559 (2003).
- Wang, Q., Ghasem-Zadeh, A., Wang, X. F., Iuliano-Burns, S. & Seeman, E. Trabecular bone of growth plate origin influences both trabecular and cortical morphology in adulthood. *J Bone Miner Res* **26**, 1577-1583, doi:10.1002/jbmr.360 (2011).
- Grüneboom, A. *et al.* A network of trans-cortical capillaries as mainstay for blood circulation in long bones. *Nat Metab* **1**, 236-250, doi:10.1038/s42255-018-0016-5 (2019).

6. Pg7 line 151, 152 comparisons of Emcn+ area with Emcn transcript levels: Emcn+ area is not statistically increased, but transcripts are.

We have changed the sentence now (page 8): Although EMCN⁺ area was not significantly elevated (**Fig. 3d**), EC-specific depletion of SMAD1/5 increased *Emcn* mRNA abundance (fold change between means 1.5; $p = 0.02$; **Fig. 4e**).

7. Pg10, there is inconsistency in the text description and data in figure 5. Please check figure 5

We like to thank the reviewer for careful reading, and we have now corrected the references to the Fig. 5.

8. On line 189, cell morphology quantification data not provided

We agree and have deleted “cell morphology”.

9. Figure 5d should show anastomosis data. No data provided. Also needs quantification.

Based on the valuable suggestions of the reviewer and as explained in the response to comment 1, we have now added the respective data on the vessel arch numbers at P35 to Fig. 3 (page 9). We have made the following changes in the text:

- Results (page 12):
However, by P35, 14 days post-tamoxifen, EC-specific SMAD1/5 depletion resulted in dysmorphogenesis of the anastomotic arches at the chondro-osseous junction (**Fig. 5f**; arrows; cf. **Fig. 3h, g**) and a significant enlargement of the hypertrophic zone (hz) of the growth plate (**Fig. 5g, h**).

10. The subtitle ‘Endothelial SMAD1/5 signaling regulates endomucin expression and vascular maturation in diaphyseal sinusoidal (type L) capillaries’ does not justify the text and data provided in this section. There is no data on vascular maturation and Emcn regulation. Did authors check Emcn levels in endothelial cells after activation of Smad1/5. Subtitle needs to be amended or supportive data has to be provided.

We agree with the reviewer and have removed “endomucin expression” from the subtitle (page 14).

11. Pg 12, line 217, give reference for sprouting angiogenesis of type L and coupling with hematopoiesis.

We have removed the “form by sprouting angiogenesis” and added the following references to support “functionally couple with hematopoiesis in the bone marrow”:

Kusumbe, A. P., Ramasamy, S. K. & Adams, R. H. Coupling of angiogenesis and osteogenesis by a specific vessel subtype in bone. *Nature* **507**, 323-328

Itkin, T. *et al.* Distinct bone marrow blood vessels differentially regulate haematopoiesis. *Nature* **532**, 323-328

12. Pg12, Vascular loops need to be explained. Should include Type h quantification see comment 3.

We have now included detailed explanations on the quantification of vascular loops – please see response to comment 4.

13. Authors mention tip/stalk cells and sprouting angiogenesis but there is no data provided that supports the claim. Authors need to provide data to support the presence of sprouting angiogenesis in type L vessels.

We value the important comment from the reviewer and have deleted the *Dll4* data and respective text referring to sprouting angiogenesis in the Result (page 14) and Discussion section (page 20).

14. Concluding sprouting angiogenesis based on total bone marrow Dll4 mRNA levels is biased. Dll4 is also expressed by hematopoietic cells in the marrow. An increase in Dll4 levels could be associated with the change in the hematopoietic compartment, which forms the majority of cell fraction in the marrow.

Please see response to comment 13.

15. Vascular permeability- bone sinusoids (type L vessels) are fenestrated and do not have barriers. So erythrocyte immunostaining does not indicate vascular permeability. Did authors check permeability by any other methods? Is there any change in erythropoiesis?

We thank the reviewer for raising this important question. The clinical manifestation of a SMAD1/5 depletion is a genetic defect in the ALK1 signaling (receptor upstream of SMAD1/5) which causes the autosomal dominant vascular disorder, hereditary hemorrhagic telangiectasia (HHT). HHT results in arteriovenous malformations (AVM) and vessel wall fragility with the risk for fatal hemorrhage in human patients which can also affect the bone marrow. Thus, it is known that SMAD1/5 depletion induces vessel permeability, and our data confirms this phenotype in the bone marrow *. We have clarified our interpretation of the results in the text and added a more comprehensive Figure in the Supplementary Information (Supplementary Fig. 6). We additionally stained for CD71 to mark erythroid progenitors (CD71+Ter119+) and found qualitatively decreased CD71 positivity at later stages. The following changes have been made:

- Results (page 14):

As clinical vascular disorders caused by genetic defects in BMP-ALK1 signaling (hereditary hemorrhagic telangiectasia; HHT) are also characterized by vessel wall fragility, we stained for Ter119⁺ erythrocytes to assess extravascular red blood cell abundance. Because erythropoiesis does occur in the bone marrow, we additionally evaluated CD71⁺ erythroid progenitors (**Supplementary Fig. 6**). We observed an increase in extravascular red blood cells, consistent with a HHT-related vascular permeability phenotype. However, we also observed qualitatively decreased CD71 positivity at later stages (P35; **Supplementary Fig. 6**). These observations suggest a potential role of vascular SMAD1/5 activity in vascular permeability and barrier function but may also indicate regulation of erythropoiesis.

• Supplementary Fig. 6:

Supplementary Figure 6. EMCN, Ter119 and CD71 staining in the diaphysis. Representative images for n= 2-7 per group. Scale bars indicate 250 μ m.

References*:

- Kulikauskas, M. R., X, S. & Bautch, V. L. The versatility and paradox of BMP signaling in endothelial cell behaviors and blood vessel function. *Cellular and Molecular Life Sciences* **79**, 77
- Willis, J., Mayo, M. J., Rogers, T. E. & Chen, W. Hereditary haemorrhagic telangiectasia involving the bone marrow and liver. *British Journal of Haematology* **145**, 150-150

16. Pg 14, line 252. High mag images and quantification data need to show 'branching and network formation' (figure 7b)

We have removed "of pronounced branching and network formation".

17. Supplementary fig1. Labels are missing - wt and cre+ images

We excuse our mistake and have now added the labels.

REVIEWERS' COMMENTS:

Reviewer #1 (Remarks to the Author):

The revised manuscript was significantly improved. Now it is easier to read and follow the results. There are still few point that have to be addressed. Please see my comments in a point-wise manner in the attached file.

Reviewer #2 (Remarks to the Author):

All of my concerns have been addressed except for my concern about the cortical porosity. The strangely high values call into question the technical quality of the imaging and image processing, which then has implications for more of the results. Simply removing the cortical porosity dataset is insufficient.

Reviewer #3 (Remarks to the Author):

Authors have carefully addressed all the comments and supported their claims with convincing data. I feel the manuscript is now ready for publication.

One minor suggestion:

In Figure 3 graph panels i and j, the distribution of data points in Smad 1/5 EC mutants indicates the need for more samples. With this data, i would be cautious about concluding 'no change'. I would either remove this graph from the paper or include more data points.

Point-by-point Response to Reviewers

Dear Dr. Rauner & anonymous reviewers:

Thank you for your handling of our manuscript, for the opportunity to address the final reviewers' feedback with this revision, and for the provisional acceptance. We have addressed the reviewers' comments with additional edits to the manuscript, and we are excited to resubmit this revision. We have noted the changes we have made below. Please find included both a copy of the manuscript with all changes tracked as well as a clean copy for easier reading. Please find the reviewers' comments in *blue* followed by our responses (page numbers refer to the copy with changes tracked) below:

Response to Reviewer #1

The revised manuscript was significantly improved. Now it is easier to read and follow the results. There are still few point that have to be addressed. Please see my comments in a point-wise manner in the attached file.

We would like to thank the reviewer for the final review, and for the suggestions. We have addressed the reviewer's comments in a point-wise manner below, indicating how the manuscript has been changed in our response.

However, the mouse femur increases its length between week 3 and 5 by 35-40% (Brylka et al., Plos one, 2017), which is quite significant. Therefore, detecting the difference in bone growth should be possible at 2 weeks post-tamoxifen if it is present.

We thank the reviewer for this comment. We did not measure or examine long bone lengthening *per se*, so cannot address this question in the current study, but this will be point we aim to address in future studies.

MMP9 is extracellular enzyme that was shown to be enriched around tip cells at the growth plate-ossification front border (Dzamukova et al., 2022, Romeo et al., 2019). On the newly provided images, the MMP9 staining seems to rather intracellular and completely missing around tip cells in the growth plate, where it should be primarily enriched. Therefore, the specificity of the shown staining is highly doubtful, and the images should be removed from the manuscript together with the corresponding conclusions.

As mentioned before, the conclusion on MMP9 results is not justified based on the provided staining. VEGFR3 expression in not assessed in tip cells, the authors should discuss that the enlarged growth plate might be a result of an impaired tip and stalk cells formation.

As both comments address the same topic, we take the liberty to address them together. Romeo et al. showed higher Mmp9 mRNA expression in type H endothelial cells using RNA sequencing after cell sorting, which suggests enriched MMP9 activity in these cells. Consistently, we recently observed regulation of Mmp9 mRNA expression in osteoblast precursors and endothelial cells during bone development, using both scRNA-seq and immunostaining (Collins, Lang et al. Developmental Cell 2024). Likewise, Dzamukova et al. stained for MMP9 at different ages and observed enrichment around tip cells, but also metaphyseal vessels in general. Notably, Dzamukova et al. used thick sections, which might explain some differences. Our staining indicates both cellular and extracellular expression of MMP9 in metaphyseal vessels which is markedly reduced in the SMAD1/5 deficient mice (P35; 2 weeks). To address the comment, we have revised the text in the Results and mentioned the alterations in tip and stalk cell formation in the Discussion, as detailed below:

Results (page 10)

*It has been shown that MMP9 activity in metaphyseal vessels directs growth plate size during bone development 26. Therefore, we performed additional MMP9 staining in the metaphysis and **found evidence***

for reduced metaphyseal MMP9 abundance by P35, 14 days post-tamoxifen (Supplementary Figure 5). Together, these data establish the necessity of ongoing SMAD1/5 signaling in maintenance of metaphyseal vessel-mediated resorption of hypertrophic cartilage and growth plate remodeling.

Discussion (page 18)

that EC-specific SMAD1/5 depletion resulted in a significant enlargement of the hypertrophic zone of the growth plate. Therefore, the dysmorphogenesis of the metaphyseal vessels, the potential altered tip and stalk cell formation and/or the reduced expression of MMP9 at the chondro-osseous junction could be the reason for the enlargement of the hypertrophic chondrocyte zone upon EC-specific SMAD1/5 depletion.

Response to Reviewer #2

All of my concerns have been addressed except for my concern about the cortical porosity. The strangely high values call into question the technical quality of the imaging and image processing, which then has implications for more of the results. Simply removing the cortical porosity dataset is insufficient.

We apologize for the unclear explanation and thank the reviewer for catching this issue. We removed these data not because of imaging inaccuracies or segmentation/threshold errors, but due to an error in the contouring, which rendered this particular parameter inaccurate. It did not affect the calculation of the other morphometric parameters. In checking the data and analysis pipeline, we recognized that the ROI/VOI definition we selected included the bone marrow channel in the calculation of cortical porosity, which explains the higher values. It would be possible to manually re-draw the contours to exclude the marrow channel; however, inspection suggests that our perturbation had no effects on cortical porosity, so we decided to focus on the standard measurements of cortical bone morphometry, as shown in the revised manuscript.

Response to Reviewer #3

In Figure 3 graph panels i and j, the distribution of data points in Smad 1/5 EC mutants indicates the need for more samples. With this data, I would be cautious about concluding 'no change'. I would either remove this graph from the paper or include more data points.

We thank the reviewer for the valuable comment. We have refined the corresponding text which reads now as following (page 7):

Differences in EMCN+ and CD31+ areas in the metaphysis were not statistically significantly, but reflect a qualitative reduction (Fig. 3i, j). Together, these data demonstrate a role of endothelial SMAD1/5 signaling in short- and long-term morphogenesis of metaphyseal vessels.